

# Satellite-based modeling of wetland methane emissions on a global scale (SatWetCH4 1.0)

Juliette Bernard[1,2], Marielle Saunois[1], Elodie Salmon[1], Philippe Ciais[1], Shushi Peng[3], Antoine Berchet[1], Penélope Serrano-Ortiz[4], Palingamoorthy Gnanamoorthy[5,6], and Joachim Jansen[7]

[1]Laboratoire des Sciences du Climat et de l'Environnement, CEA-CNRS-UVSQ, Gif-sur-Yvette, France
[2]LERMA, Paris Observatory, CNRS, PSL, Paris, France
[3]College of Urban and Environmental Sciences, Peking University, Beijing 100871, China
[4]Department of Ecology, Andalusian Institute for Earth System Research (CEAMA-IISTA), University of Granada, Spain
[5]CAS Key Laboratory of Tropical Forest Ecology, Xishuangbanna Tropical Botanical Garden, Chinese Academy of Sciences, Menglun, China
[6]Coastal Systems Research, M. S. Swaminathan Research Foundation, Chennai, India
[7]Department of Ecology and Genetics/Limnology, Uppsala University, Uppsala, Sweden

**Correspondence:** Juliette Bernard (juliette.bernard@obspm.fr)

**Abstract.** Wetlands are major contributors to global methane emissions. However, their budget and temporal variability remain subject to large uncertainties. This study develops the Satellite-based Wetland CH$_4$ model (SatWetCH4), which simulates global wetland methane emissions at 0.25°x0.25° and monthly temporal resolution, relying mainly on remote sensing products. In particular, a new approach is derived to assess the substrate availability, based on Moderate-Resolution Imaging Spectrora-
diometer data. The model is calibrated using eddy covariance flux data from 58 sites, allowing for independence from other estimates. At the site level, the model effectively reproduces the magnitude and seasonality of the fluxes in the boreal and temperate regions, but shows limitations in capturing the seasonality of tropical sites. Despite its simplicity, the model provides global simulations over decades and produces consistent spatial patterns and seasonal variations comparable to more complex Land Surface Models. In addition, our study highlights uncertainties and issues in wetland extent datasets and the
need for new seamless satellite-based wetland extent products. In the future, there is potential to integrate this one-step model into atmospheric inversion frameworks, thereby allowing optimization of the model parameters using atmospheric methane concentrations as constraints, and hopefully better estimates of wetland emissions.

## 1 Introduction

The article 1.1 of the Ramsar Convention (1971) defines wetlands as "areas of marsh, fen, peatland, or water, whether natural
or artificial, permanent or temporary, with water that is static or flowing, fresh, brackish, or salt, including marine water areas the depth of which at low tide does not exceed six meters". Each wetland exhibits very specific local conditions, such as water source (ombrotrophic or minerotrophic source) and quantity (groundwater level, soil moisture), vegetation (types, density), and soil properties (pH, carbon content, microbial communities). These areas harbor a rich biodiversity of flora and fauna and play a significant role in regulating water resources, water purification, and flood prevention (Denny, 1994; Meli et al., 2014).



Wetlands are also a crucial element for climate. On the one hand, waterlogged conditions in wetlands lead to a reduction in the rate of decomposition of soil organic carbon (SOC) and thus to a significant accumulation of carbon, such as in peatlands. This wetland SOC stock has been estimated at around 520 to 710 PgC worldwide (Poulter et al., 2021). On the other hand, anaerobic conditions favor the production of methane (Torres-Alvarado et al., 2005), a powerful greenhouse gas with a global warming potential of 80 ±26 over 20 years (IPCC 2021 AR6 Chap.7, Table 7.15). In the last Global Methane Budget (GMB)

(Saunois et al., 2020), it has been estimated that methane emissions from wetlands contribute for approximately 12 to 36% of the total methane sources. These estimates have been established from bottom-up (102-182 Tg $CH_4$ $yr^{-1}$, 12-31% of total annual sources) and top-down approaches (159-200 Tg $CH_4$ $yr^{-1}$, 27-36% of total annual sources).

Top-down approaches rely on a prior estimate of the ensemble of methane fluxes, including prior knowledge of wetland emissions, and are therefore dependent on bottom-up estimates. Bottom-up approaches estimate methane fluxes from wetlands

using formulations ranging from the simplest to the most complex, such as in Land Surface Models (LSMs). LSMs represent the budgets of water, energy, and carbon under some meteorological constraints. They account for soil processes in a series of successive steps that explicitly simulate part or all of the following processes: methane production, oxidation, and transport by diffusion, ebullition or higher plants (Riley et al., 2011; Morel et al., 2019; Salmon et al., 2022).

In the context of climate change, understanding past and predicting future trends in global wetland methane emissions are

key issues, but these trends are still uncertain (Jackson et al., 2020). Although they try to represent complex pathways involved in methane emissions, LSMs models still lead to significant uncertainties in terms of global total emissions, seasonal cycle and spatial patterns (Melton et al., 2013; Saunois et al., 2020). In particular, the internal wetland surface area varies considerably from one LSM to another (Melton et al., 2013). Moreover, a large part of the studies (Zhu et al., 2013; Bohn et al., 2015; Guimberteau et al., 2018; Peltola et al., 2019; Qiu et al., 2019; Salmon et al., 2022; Tenkanen et al., 2021; Kuhn et al., 2021;

Rößger et al., 2022) focus only on boreal and temperate regions. In fact, the boreal regions are of great interest because temperatures there are rising faster than the global average (England et al., 2021; Post et al., 2019; Previdi et al., 2021) and permafrost is thawing, which could lead to large increases in carbon dioxide and methane emissions (Schuur et al., 2022). However, about three quarters of global wetland methane emissions actually occur in tropical regions (Saunois et al., 2020), where wetland methane emissions are still poorly understood (Meng et al., 2015), partly due to the scarcity of measurements

in tropical wetlands compared to boreal and temperate regions (Delwiche et al., 2021).

Simpler formulations than LSMs, operating on a global scale (Gedney, 2004; Bloom et al., 2017; Albuhaisi et al., 2023) implicitly represent soil processes in a one-step approach between soil organic carbon content, which is the main substrate for methanogenesis, and $CH_4$ emissions. While these models may not provide greater accuracy compared to LSMs, they have the advantage of operating faster (within a few seconds) and relying on only a few parameters and variables. They provide quick

estimates and can be valuable for sensitivity testing or trend analysis. Typically, the variables considered in the different models are the wetland area, the soil temperature, a proxy for carbon substrate and sometimes a local water variable (water table depth, WTD, or soil water content, SWC). The differences between these simple models depend on the equation formulation, the choice of data sets used to constrain the variables and the calibration method.





Methanogenic bacteria use organic carbon from litterfall, root exudates, dead plants and dissolved organic carbon that has
already been broken down to low molecular weight molecules by other microorganisms (Nzotungicimpaye et al., 2021; Torres-
Alvarado et al., 2005; Bridgham et al., 2013). Quantifying the organic matter available for methanogenesis is not trivial, as
it cannot be measured directly. Many proxies are used in the literature without a consensus being found (Wania et al., 2013;
Melton et al., 2013): Some models use NPP as a proxy (e.g. UW-Vic, Walter and Heimann (2000)), while others consider
that methane production could be derived by multiplying heterotrophic respiration by a $CO_2$:$CH_4$ ratio (e.g. LPJ, CLM4Me,
SDGVM). Other models use SOC as a proxy for carbon available for methanogenesis (Gedney, 2004). However, not all SOC
can be used for respiration by methanogenic bacteria. Carbon pool models are embedded in some LSMs such as ORCHIDEE
(Ringeval et al., 2010; Salmon et al., 2022) to distinguish readily available SOC from recalcitrant SOC.

In the absence of global data on substrate availability, Gedney (2004) proposed a simple equation based on wetland fraction,
temperature, and total soil carbon. These three variables were modelled using the Met Office climate model (Gordon et al.,
2000) coupled to the land surface scheme MOSES-LSH (Gedney and Cox, 2003), and their model was run for the period 1990-
1998. Bloom et al. (2017) also used a simplified approach based on an equation relying on wetland fraction, soil temperature,
soil heterotrophic respiration, and fed with different datasets, forming the WetCHARTs 1.0 ensemble for 2001-2015. The
heterotrophic respiration data were derived by terrestrial biosphere models. In general, the proxies used in these studies are
derived from models (LSMs, hydrological models...) and in some rare cases from remote sensing data. Recently, Albuhaisi
et al. (2023) proposed a methane emission formulation fed only by satellite and satellite-derived data sets for soil moisture and
SOC. However, this approach was carried out only in the boreal region for the period 2015-2021.

The calibration methods of these approaches have varied in recent years due to important changes in the available data.
The first attempt by Gedney (2004) assumed that the global atmospheric concentration anomalies were solely due to wetlands.
This approximation is highly questionable according to current estimates of anthropogenic and natural methane emission
trends (Jackson et al., 2020). Too few flux data measurements were available in the early 2000s to be used for calibration. In
WetCHARTs (Bloom et al., 2017), the model calibration was performed by constraining total wetland methane emissions to
the GMB ensemble mean (Saunois et al., 2016), and as such was not independent of other LSM approaches. However, recent
efforts by the FLUXNET community (Delwiche et al., 2021) have led to the construction of a unified database of methane
fluxes measured by eddy covariance worldwide, offering the possibility of new independent calibration methods. The eddy
covariance method provides stable and continuous in situ flux measurements over relatively large areas (>100m$^2$) with limited
environmental disturbance (Baldocchi et al., 2001; Kumar et al., 2017). The FLUXNET-CH4 database includes some ancillary
data such as soil temperature, gross primary productivity, WTD or SWC, but not for all sites. An important issue is still the
inhomogeneous distribution of flux towers across the globe, with sites mainly located in temperate and boreal regions. Albuhaisi
et al. (2023) used 12 flux stations available between 2015 and 2018 from the FLUXNET-CH4 database to calibrate the scaling
parameter of their boreal emission models, but the two other parameters ($Q_{10}$ and $T_0$) were set according to literature values.

In addition to this improvement in available methane flux data, new dynamic estimates of wetland area have emerged since
the studies of Gedney (2004) and Bloom et al. (2017). These estimates are based on either satellite observations or hydrological
models. WAD2M, published by Zhang et al. (2021a), provides a complete dynamic map of wetlands, including peatlands. It





is partly based on satellite data and is widely used in the community, especially for the GMB (Saunois et al., 2020). Xi et al.
(2022) produced an ensemble of 28 wetland extent products derived from TOPMODEL, a hydrological model.

Our study aims to revise the simplified modelling approach for wetland methane emissions proposed by Gedney (2004), taking advantage of recent developments. The objective is to develop a model framework capable of reproducing the main features of wetland methane emissions (annual budget, seasonal cycle, spatial distribution) on a global scale with a resolution of 0.25°x0.25°. The Satellite-based Wetland CH$_4$ model (SatWetCH4) is based on a data-driven approach, mostly fed with

satellite-derived datasets, to allow fast and easy sensitivity calculations. SatWetCH4 provides an independent estimate, and uses in situ eddy covariance data for model calibration. Particular attention has been paid to the proxy for available carbon. As methanogenic activity has been shown to be related to plant productivity (Bridgham et al., 2013), here we use a MODIS plant photosynthesis product to derive a $C_{substrate}$ dataset to assess the organic matter available for methanogenesis, as described in section 2.1. The aim of deriving the $C_{substrate}$ product is to obtain a carbon product that 1) best represents the carbon available

for methanogenesis, 2) is dynamic, 3) is based on satellite data, and 4) is independent of LSMs.

Section 2 presents the materials and methods, including the model, the satellite-based input datasets and the calibration procedure. Optimization results are presented in section 3.1, followed by a site-level evaluation of the model in section 3.2. The global-scale results for the period 2003-2020 are presented in section 3.3. Section 4 examines the model's limitations and prospects for improvement given the current state of modeling.

## 2 Materials and Methods

### 2.1 Model description

We estimate the methane flux using the following formulation, similar to that of Gedney (2004):

$$F_{CH_4} = k f_w C_{substrate} Q_{10}(T)^{(T-T_0)/10} \tag{1}$$

where $k$ is a scaling factor, $f_w$ the wetland fraction of the pixel, $C_{substrate}$ the carbon content that is available for methanogen-

esis, and $T$ the soil temperature. $Q_{10}(T)$ depends on $Q_{10}^0$ the temperature sensitivity of methanogenesis and $T$. It is defined by $Q_{10}(T) = Q_{10}^0{}^{T^0/T}$. $T^0$ is set to $273.15°K$, resulting in low emissions for frozen or near frozen soils. Consequently, $Q_{10}^0$ and $k$ are the two parameters to be calibrated.

The substrate available in the soil for methanogenesis, $C_{substrate}$, is calculated independently, upstream of the model. It is constructed as a litter pool model scheme and depends on temperature, Net Primary Productivity (NPP) and varies with time.

This $C_{substrate}$ is computed using the following equation :

$$\frac{dC_{substrate}}{dt} = NPP - K(T)C_{substrate} \tag{2}$$

In this scheme, the available substrate is assumed to originate mainly from photosynthesis, which is approximated as NPP. The second term represents the carbon loss due to soil heterotrophic respiration, which depends on a turnover rate function





$K(T) = K^{ref} Q_{10K}^{(T-T_K^{ref})/10}$. $K^{ref}$ reflects the reference turnover time, $Q_{10K}$ the temperature sensitivity coefficient of respi-

ration, and $T_K^{ref}$ the reference temperature. Incubation experiments (Parton et al., 1987; Khvorostyanov et al., 2008; Schädel et al., 2014) provided estimates of $K$ between 0.2 and 2.5 years, corresponding to a residence time of carbon in soils between 0.4 and 5.5 years. Therefore, to obtain a consistent $K$, the model parameters are set to $K^{ref} = 1/2yr = 0.5yr^{-1}$, $T_K^{ref} = 303.15°K$, and $Q_{10K} = 2$.

The global estimate of $C_{substrate}$ is established in advance by discretizing Eq.2 at monthly time steps. The $C_{substrate}$ was

primarily run for 100 years to reach an equilibrium stage, constrained with 2001 NPP values obtained from remote sensing data (Zhang et al., 2017) detailed in section 2.3. NPP data between 2003 and 2020 were then used to estimate $C_{substrate}$ over the same period on a monthly scale.

## 2.2    In situ data

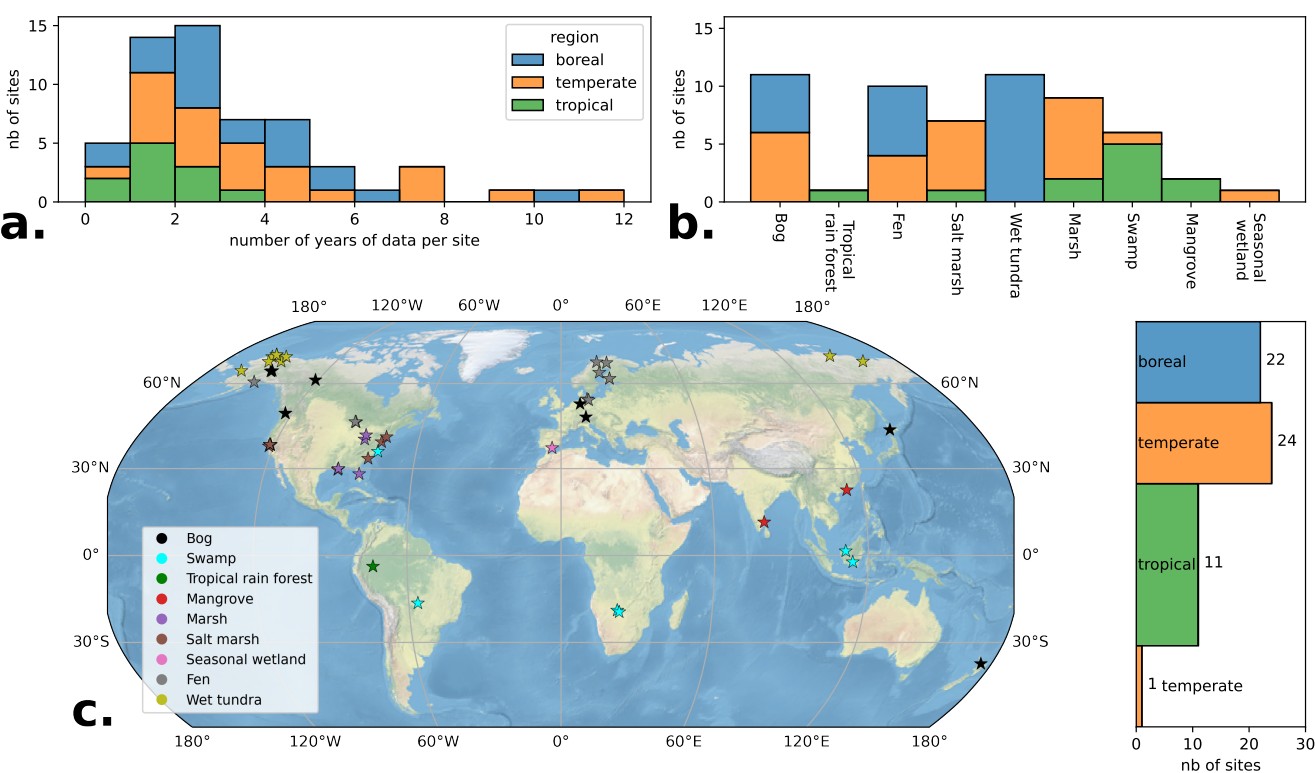

**Figure 1.** Sites distribution per **a.** length of available observation period, **b.** wetland type and **c.** geographic location. In map **c.**, because of their close location (few km) some sites overlaps. Site color depends **a.** and **b.** on site latitude : boreal (55°N-90°N), temperate (30-55°N or °S), or tropical (30°S-30°N), and **c.** on wetland type.





Eddy covariance time series of methane fluxes from different databases were combined in order to use robust, continuous
and the longest methane flux monitoring period recorded at each site. In situ data from 58 wetland sites were collected from
FLUXNET-CH4 (Delwiche et al., 2021), AmeriFlux (Baldocchi et al., 2001), EuroFlux (Valentini, 2003). In addition, data
for BW-Gum and BW-Npw sites were obtained from the UK Centre for Ecology & Hydrology website, and IN-Pic data were
provided through personal exchanges with the principal investigator, P. Gnanamoorthy. Some ancillary variables of interest for
methane emission modelling (e.g. soil temperatures, WTD, SWC, precipitation) are available at some of the sites. Links to the
sources used are given in Supplementary Table S1 and the full list of sites and details are listed in Supplementary Table S2.

The length of the time series, wetland types, and location of the sites are presented in Fig.1. Despite the construction of
the most comprehensive database from recent literature, the global distribution of methane eddy covariance tower sites shows
significant heterogeneity. The majority of sites, 46, are located at latitudes greater than 30°N, with 36 sites in North America
and 10 sites in Europe. Only 11 sites (19%) are located in the tropical band 30°S-30°N, including, for example, only 2 sites
on the entire African continent, which are only a few kilometers apart, and 2 sites in South America. In addition, Fig.1.a
highlights the heterogeneity in measurement duration, with tropical sites having a median measurement duration of 1.6 years,
as contrasted with 2.7 and 3.2 years for boreal and temperate sites, respectively. It is also important to note that sites can be
very close to each other (within a few kilometers). This uneven distribution of sites introduces a bias in the global calibration
of the model. In particular, tropical wetlands are severely underrepresented, although they are expected to account for about
~75% of global wetland methane emissions (Saunois et al., 2020).

To ensure a homogeneous dataset, the same data processing was applied to the raw data. The 30 min raw data points
were extracted, and the variable units were unified. Outliers are removed for all variables, including ancillary data, notably
for methane fluxes, for each site and day, data outside of $\overline{F_{CH4day}} \pm 5std_{F_{CH_4}day}$ are excluded. Finally, daily averages are
calculated for all variables, and monthly averages are only calculated if more than 4 days of data are available in a given
month. A monthly time scale was chosen because it effectively captures seasonal variations while minimizing the influence of
variables that operate at shorter time intervals, such as daily or multi-day changes in atmospheric pressure, or diurnal cycles in
vegetation and temperature (Knox et al., 2021). Furthermore, as our model is a one-step model without differentiation between
production and emissions, the monthly time scale also mitigates potential errors due to time lags between methane production
and transport (Ueyama et al., 2023).

This results in a dataset of 2354 monthly mean methane fluxes associated to their available ancillary data.

## 2.3 Global forcing datasets

### 2.3.1 MODIS PSnet data

To derive $C_{substrate}$ estimates, as defined in Eq.2 in section 2.1, we use PsnNet from the MODIS MOD17A2HGF v6.1 dataset
(Running and Zhao, 2021). The PsnNet dataset represents NPP, except that it excludes growth and maintenance respiration
costs. This product is based on satellite Fraction of Photosynthetically Active Radiation (FPAR) data, a reanalysis meteorolog-
ical dataset, and land cover classification.



The data cover the period from 2000 to the current year, but data for 2002 are not available, so only the period from 2003 to 2020 has been used in this study. The PsnNet product has been regridded from the native 500 m resolution to a 0.05° product used for model optimization at the site level, and to a 0.25° resolution product used for the global simulation. In terms of timescale, monthly averages were estimated from the initial 8-day product.

### 2.3.2 ERA5 soil temperature

For the soil temperature variable, monthly averaged data from ERA5-Land, available at https://cds.climate.copernicus.eu/, are used. The temperature in the 7-28 cm soil layer is selected, denoted as *lay2*. These data are available from 1950 to the present with a resolution of 0.1°x0.1°. A comparison of in situ soil temperature measurements with ERA5 land *lay2* closest 0.1° pixel is detailed in the Supplementary Fig.S1, showing good agreement between in situ and ERA5 soil temperatures, with in particular a high temporal correlation (r>0.9) and low RMSD (<2°K) for 37 of the 42 sites equipped with temperature probes.

### 2.3.3 Global wetland extent datasets

Two wetland areas are used to estimate global methane emissions. The Wetland Area and Dynamics for Methane Modeling (WAD2M) version 2.0 (Zhang et al., 2021a) describes the fraction of wetlands per pixel globally at a resolution of 0.25°x0.25° for the period 2000-2018 with a monthly time step. The dynamics of WAD2M are driven by the Surface Water Microwave Products Series (SWAMPS) (Jensen and Mcdonald, 2019), which relies on passive and active microwave satellite observations. Several static datasets are used to add non-inundated wetlands, such as peatlands, and to remove lakes, irrigated rice paddies (Zhang et al., 2021b). The second wetland map used is based on the TOPography-based hydrological MODEL (TOPMODEL). Xi et al. (2022) built an ensemble of 28 maps describing globally the fraction of wetlands per pixel at a resolution of 0.25°x0.25° for the period 1980-2020 at a monthly time step (Xi et al., 2021). A combination of 7 different soil moisture reanalysis datasets and 4 different surface wetland extent products were used to calibrate the model. Among the 28 products, we select here the version calibrated with ERA5 soil moisture data and the GIEMS-2 (Prigent et al., 2020) long-term maximum, as it shows the highest correlations of wetland area with the original wetland product (Xi et al., 2022).

### 2.4 Calibration method

The in situ methane fluxes at the sites were used to calibrate the SatWetCH4 model parameters $k$ and $Q_{10}^0$. Model calibration at site level implies that each site is considered to be completely covered by wetland, resulting in a wetland fraction of 1 ($f_w = 1$). The flux equation to be optimized at site level is then $F_{CH_4} = kC_{substrate}Q_{10}(T)^{(T-T_0)/10}$. The $C_{substrate}$ product (described in section 2.1) and ERA5 soil temperature (described in section 2.3) are used as input variables by selecting the nearest pixels to the sites, at 0.05° for $C_{substrate}$ and 0.1° for ERA5 soil temperature respectively.

Least squares regression is performed simultaneously on all sites using the Broyden-Fletcher-Goldfarb-Shanno algorithm (Byrd et al., 1995). For sites with less than 12 months of data, a weight proportional to the number of monthly measurements is assigned to the site data. Sites with more than 12 months of data are given equal weights. The minimized cost function is :





$$J = \sum_{sites} w_{site} MSD_{site} = \sum_{sites} w_{site} \overline{(F_{CH_4obs} - F_{CH_4sim})^2_{site}} \tag{3}$$

where $w_{site}$ is the site weight, $MSD$ is the Mean Square Deviation, $F_{CH_4obs}$ is the in situ methane flux observed at the sites, and $F_{CH_4sim}$ is the methane fluxes simulated by the model. If the number of monthly methane flux measurements at the site, $n_{site}$, is greater than or equal to 12, $w_{site} = 1$ otherwise $w_{site} = \frac{n_{site}}{12}$. Different initial parameter sets for $k_{firstguess}$ (0.01, 0.1, 1, and 10) and $Q^0_{10firstguess}$ (1.5, 2.5, 3, and 4) are tested to evaluate the influence of the calibration initialization and to ensure the global nature of the found minimum.

## 3 Results

### 3.1 Optimized model parameters

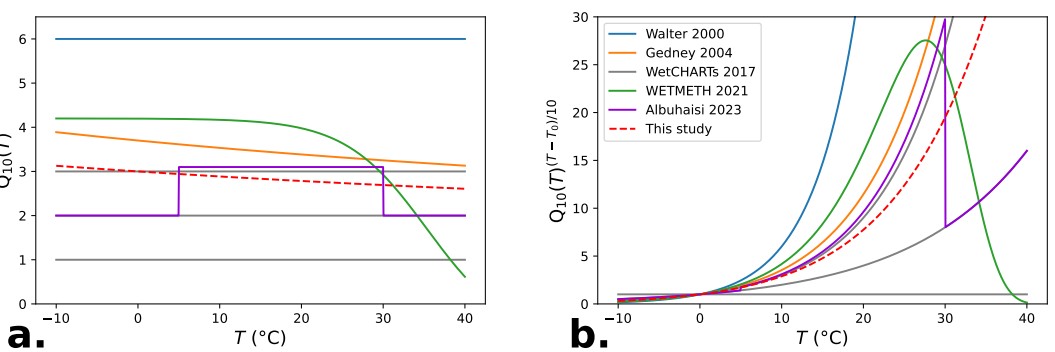

**Figure 2. a.** Comparison of $Q_{10}(T)$ formulation with Walter and Heimann (2000), Gedney (2004), WetCHARTs Bloom et al. (2017), WETMETH Nzotungicimpaye et al. (2021), and Albuhaisi et al. (2023). **b.** Effect of the different $Q_{10}(T)$ formulations when incorporated in the temperature dependency function.

The calibration is performed according to the method described in section 2.4. The minimum cost function is found for $Q^0_{10,opt} = 2.99$ and $k_{opt} = 3.097 \, 10^{-2} \, \mu gCH4/m^2/s$. The value of $k_{opt}$ has no numerical meaning, as it is highly dependent on the units and order of magnitude of the substrate proxy we use. The $Q_{10}(T)$ formulation obtained from this calibration is compared with the literature values in Fig.2a. Fig.2.b shows the influence of $Q_{10}(T)$ expressions when inserted in the temperature formulation $Q_{10}(T)^{(T-T_0)/10}$.

Nzotungicimpaye et al. (2021) in WETMETH proposed a $Q_{10}(T)$ formulation such that, when incorporated into the equation $Q_{10}(T)^{(T-T_0)/10}$, it indicates an optimal temperature range for methanogenesis around 25-30°C. Although we attempted a similar approach to formulate $Q_{10}(T)$, it resulted in minimal changes in the flux outcomes while increasing the complexity of the formulation and hindering the convergence of the cost function. Albuhaisi et al. (2023) used a reduced $Q_{10}$ for temperatures





above 5°C or above 30°C, resulting in abrupt transitions at these temperature thresholds. However, this implementation may not be appropriate for global analyses, as tropical wetlands experience temperatures above 30°C, and such sudden changes do not reflect of physical reality. Therefore, the Gedney (2004) formulation $Q_{10}(T) = Q_{10,opt}^0{}^{T_0/T}$ was used for calibration, resulting in $Q_{10}(T)$ from 3.12 (-10°C) to 2.60 (40°C), which is slightly lower than the Gedney (2004) value (3.89 at -10°C to 3.13 at 40°C). Our $Q_{10}(T)$ value contrasts with that of Walter and Heimann (2000) ($Q_{10} = 6.0$, no temperature dependence), but closely matches with the upper value used in the WetCHARTs ensemble by Bloom et al. (2017) ($Q_{10} = 3.0$, no temperature dependence) and the value chosen by Albuhaisi et al. (2023) for the 5°C-30°C range ($Q_{10} = 3.1$ for T between 5°C and 30°C, $Q_{10} = 2.0$ below 5°C or above 30°C). Consequently, similar $Q_{10}(T)^{(T-T_0)/10}$ curves are observed in Fig.2.b between our estimate and those of Gedney (2004), Bloom et al. (2017), and the 5-30°C range of Albuhaisi et al. (2023), although our formulation exhibits slightly lower values. This would result in a slightly lower increase in methane fluxes with soil temperature.

## 3.2 Evaluation of the model performance at site scale

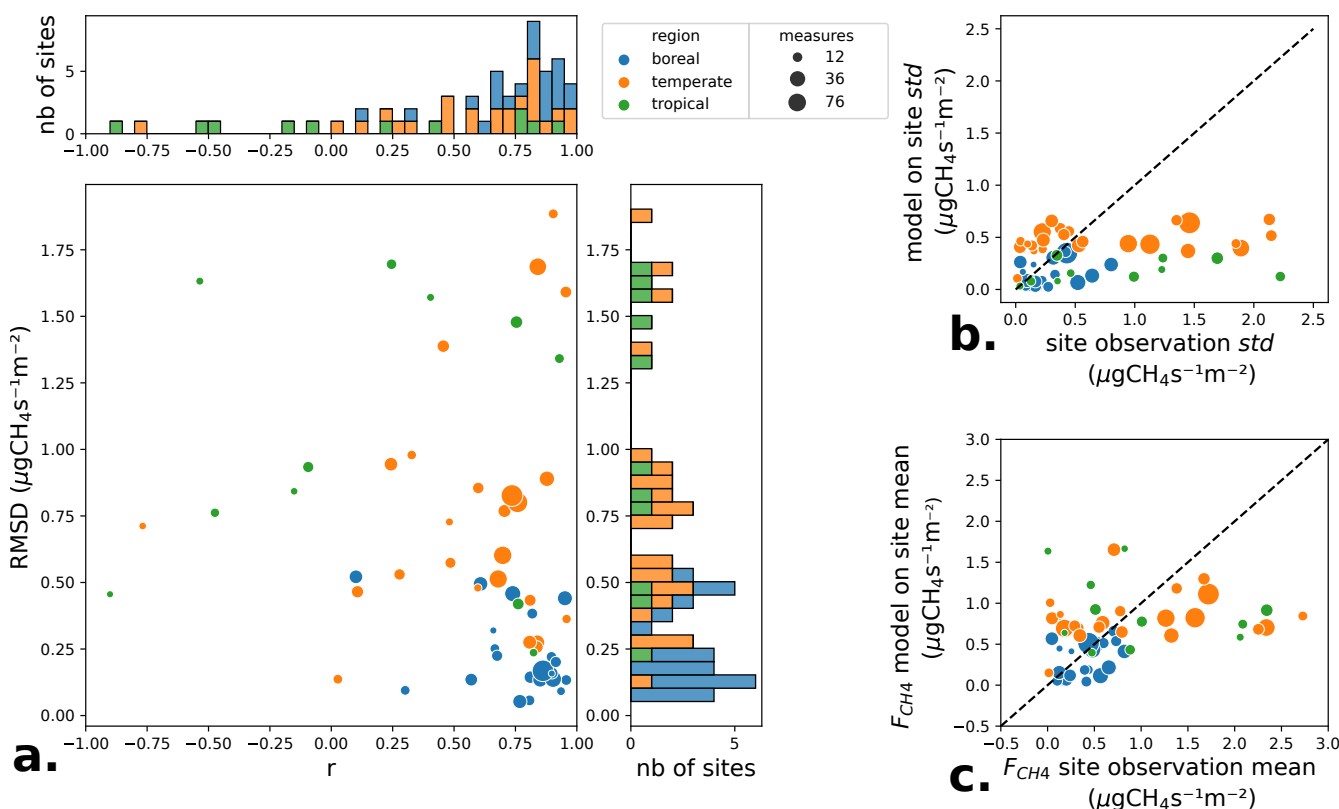

**Figure 3.** Comparison of fluxes modeled at site level with observations. Each site is represented by a point, its location by its color, while site number of measurements are represented by point sizes. **a.** Temporal correlation (r) and RMSD between model and observation. **b.** Standard deviation (std) of model fluxes in function of standard deviation of observation. **c.** Mean of model fluxes in function of mean of observation.



To evaluate the SatWetCH4 model, we run it at the site scale with the optimized parameters, setting $f_w = 1$ in Eq.1, and using the variables values from the pixels closest to the site, i.e. at 0.05° for $C_{substrate}$ and at 0.1° resolution for ERA 5 temperature. Note that the difference in spatial resolution between the site level, i.e. the footprint of the flux towers (up to 1km²), and the resolution of the available substrate (0.05°x0.05° ~25km²) limits the comparison. The temperature is more homogeneous and

its aggregation at 0.1° is less problematic. Fig. 3 compares the in situ flux data with the modeled site-level output. Fig. 3.a. shows the Root Mean Square Deviation (RMSD) and the temporal correlation (r) between the observations and the simulated flux. It indicates a generally lower average RMSD in the boreal zones (average RMSD of 0.23 $\mu$gCH4s$^{-1}$m$^{-2}$) compared to the temperate zones (average RMSD of 0.8 $\mu$gCH4s$^{-1}$m$^{-2}$) and the tropics (average RMSD of 1.1 $\mu$gCH4s$^{-1}$m$^{-2}$). It shows that the model captures the seasonality of emissions well for boreal sites (r > 0.7 for 16/22 boreal sites), less well for temperate

sites (r > 0.7 for 11/25 sites) and poorly for tropical sites (r > 0.7 for 4/11 sites, with 5/11 sites having r < 0). Fig.3.b and Fig.3.c display the amplitude variations (standard deviation) and mean values of the observed and modelled fluxes. The mean fluxes are consistent with the in situ values (Fig.3.c), while the standard deviation (std), which represents the amplitude of the seasonal variation, is underestimated for fluxes with std greater than 1 $\mu$gCH4s$^{-1}$m$^{-2}$ (Fig.3.b).

    Thus, the model reproduces boreal fluxes better than temperate and tropical fluxes. This results in higher RMSD values for

tropical and temperate zones, as shown in Fig.3.c, although these higher RMSD values are also due to generally larger fluxes in the tropics. The underestimation of fluxes in the tropics is partly due to the sampling bias mentioned in section 2.2: only a small proportion (19%) of the sites are located between 30°S and 30°N, and they have shorter monitoring periods, resulting in a cumulative weight of 18.5% in the cost function J (boreal sites weight 38% and temperate sites 43%).

### 3.3    Methane emissions from wetlands on a global scale

After calibrating $k_{opt}$ and $Q^0_{10,opt}$, we run the SatWetCH4 model (Eq.1) on a global scale for the period 2003-2020 at a resolution of 0.25°x0.25° with forcing datasets $C_{substrate}$, ERA 5 soil temperature and either WAD2M or TOPMODEL product for wetland extent at the same resolution. In the following, we compare the wetland emissions derived from SatWetCH4 in terms of total global methane emissions, spatial distribution and temporal variations with Bloom et al. (2017) and the ensemble mean of the GMB (Saunois et al., 2020).





### 3.3.1 Comparison of the spatial distribution of the wetland extents

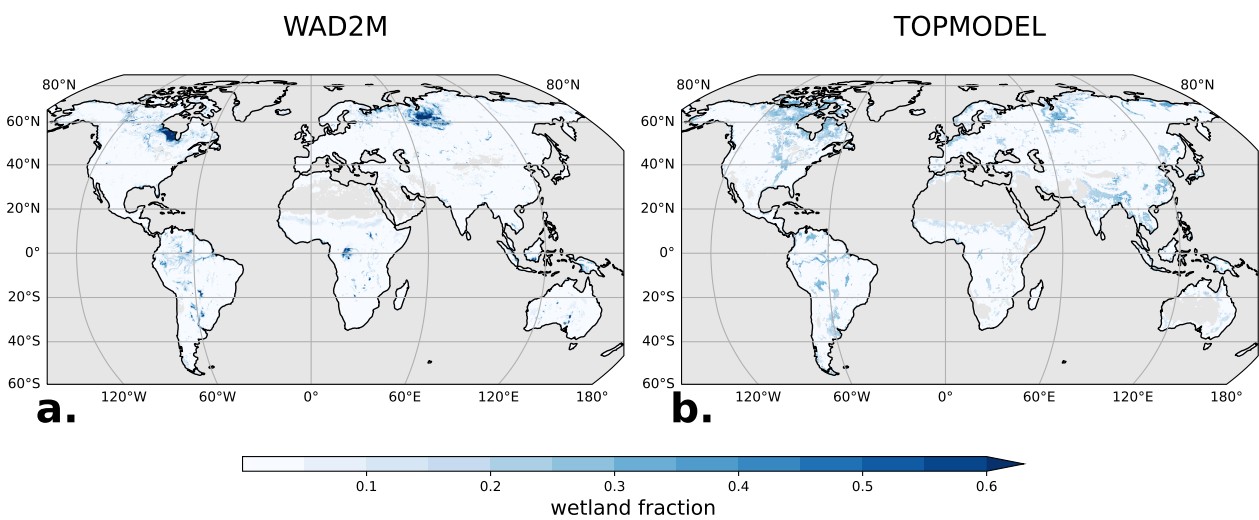

**Figure 4.** Wetland fraction mean annual mean of **a.** WAD2M and **b.** the model TOPMODEL.

For both products, the monthly average of surface extent served to derive a Mean Annual Mean (MAmean) and Mean Annual Maximum (MAmax) by selecting for each 0.25° pixel the mean or maximum of the typical 12-month seasonality. The maps of MAmean of both wetland extent products are presented in Fig.4. WAD2M has a global MAmean of 4.21 Mkm$^2$ and a MAmax of 6.76 Mkm$^2$ over 2003-2020, while TOPMODEL is lower with 3.04 Mkm$^2$ and 5.12 Mkm$^2$ respectively.

This discrepancy in the value of the total area is mainly due to the methodology employed to construct the products. First, WAD2M is known to overestimate coastal areas due to ocean contamination by nearby ocean pixels in the original SWAMPS data (Pham-Duc et al., 2017; Bernard et al., 2024). Second, WAD2M includes non-inundated wetlands, such as peatlands, whereas TOPMODEL represents only inundated wetlands. Indeed, Xu et al. (2018) estimate that peatlands cover around 4.23 Mkm$^2$. In fact, WAD2M wetland fraction over peatland areas (e.g. Hudson Bay, Congo, Siberian lowlands, Amazon floodplain) is larger than in TOPMODEL (Fig.4). Note that some boreal peatlands in WAD2M are masked by snow cover in winter, which explains the lower MAmean than the global peatland extent. There are other large spatial differences between the two datasets. Of concern in WAD2M is the substantial detection of water over Australia, a predominantly desert region. Wetlands and deserts have similar microwave signatures, explaining the possible confusion (Pham-Duc et al., 2017). Finally, TOPMODEL shows higher scattered extents over North America, India, and China than WAD2M.





### 3.3.2  $C_{substrate}$ spatial distribution

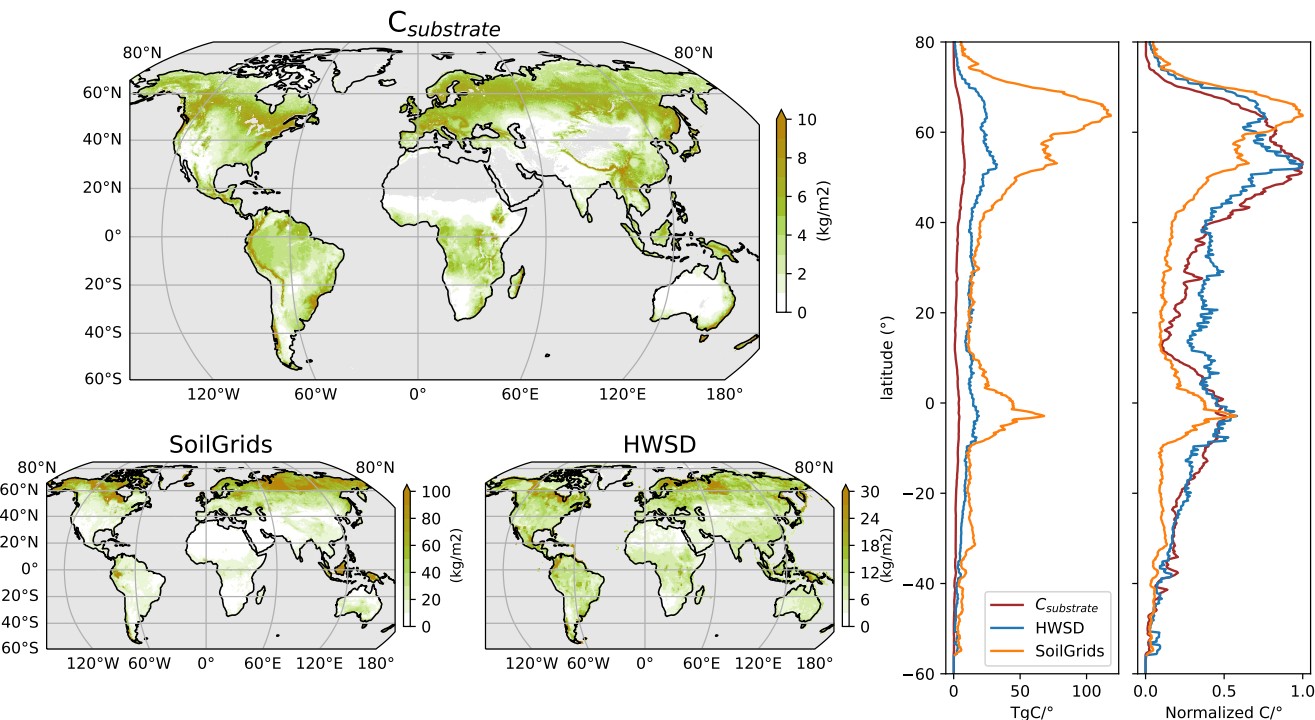

**Figure 5.** 2003-2020 mean of the derived $C_{substrate}$ product (left) and its latitudinal profile (middle) and the latitude profile normalized by the latitudinal maximum (right) along with two SOC databases for the 0-100 cm layer: HWSD (Wieder, 2014) and Soilgrids (Hengl et al., 2017).

The 2003-2020 mean map of the $C_{substrate}$ product is shown in Fig.5. This product is used as a representation of the soil carbon substrate available for methanogenesis. It should be noted that there are no analogous products for evaluation. We suggest a comparison with global estimates of 0-100cm SOC stocks derived from the World Soil Database (HWSD) (Wieder, 2014) and SoilGrids (Hengl et al., 2017) to see differences between our proxy for available substrate compared to total organic carbon

stocks. The latitudinal distribution and the latitudinal distribution normalized by the latitudinal maximum of the three products are shown on the right side of the figure.

The numerical values of $C_{substrate}$ tend to be consistently lower than those of the SOC estimates, differing by about an order of magnitude. This observation aligns with the fact that elevated SOC values, which are particularly common in peatlands, do not translate into a proportionally increased production of $CO_2$ or $CH_4$ emissions. In fact, the slow decomposition of organic

matter in peatlands leads to carbon sequestration in soils over millennia (Clymo et al., 1998). It is important to emphasize that the order of magnitude of the numerical value of $C_{substrate}$ is of limited significance, since the calibration of the $k$ factor is





used for the methane flux calculation. The critical focus is on the spatial variations and temporal dynamics of $C_{substrate}$ for accurate methane flux assessments.

The $C_{substrate}$ product shows a small seasonal variation (about 5% at global and basin scales), implying that its contribution
is mainly of spatial nature. Indeed, we observe a different spatial distribution between the three products. SoilGrids and HSWD tend to show more localized high carbon values in regions where peatlands are abundant, such as the western Siberian lowlands or the northern part of America, or for SoilGrids in Indonesia. $C_{substrate}$ presents a more homogeneous distribution, with moderate values in boreal and temperate regions. It consistently shows no or low available substrate values over bare soil regions (Sahara, Australia). In light of these considerations, $C_{substrate}$ appears to be a valuable candidate for estimating soil
carbon availability.




### 3.3.3 Spatial variations of methane emissions



**Figure 6.** SatWetCH4 modeled mean methane emission using **a.** WAD2M and **b.** TOPMODEL wetland surfaces. SatWetCH4 emissions obtained using a uniform substrate — i.e. $C_{substrate} = 1$ — and **c.** WAD2M or **d.** TOPMODEL. Emissions obtained by **e.** the mean of GMB diagnstic models and **f.** the mean of WetCHARTs ensemble.

The methane fluxes derived from SatWetCH4 are strongly influenced by the spatial patterns of the wetland extent used: the differences between WAD2M and TOPMODEL mentioned in section 3.3.1 are partly reflected in the output fluxes (Fig.6a.



and b.). In fact, the parameter $f_w$ is directly a multiplicative coefficient in the flux calculation in Eq.1. In particular, peatland

regions emit more in the WAD2M version, and the Ganges and Yangtze basins show much more intense methane emissions when TOPMODEL is used.

We assess the sensitivity of SatWetCH4 model to the $C_{substrate}$ product derived from the NPP (Eq.2) by comparing the results from SatWetCH4 reference run (with $C_{substrate}$) and a run that considers a uniform substrate ($C_{substrate}$=1 over the globe). Note that to do this, we had to calibrate the model parameters $k$ and $Q_{10}^0$ using the same method described in section

2.4. The found $Q_{10,opt}^0$ is lower (1.83 instead of 2.99). Fig.6.c and d show average emissions assuming a uniform substrate for methanogenesis, run with WAD2M or TOPMODEL. The spatial distribution is then very different, depending only on the wetland extent dataset and weighted by temperature. In particular, emissions are significantly higher in subequatorial Africa with both wetland datasets when no substrate product is included in the model. In fact, $C_{substrate}$ is small over this region due to a small value of the MODIS PsnNet input (Fig.5). Over Australia we observe significantly higher fluxes with WAD2M when

$C_{substrate}$ is not considered. This shows an overestimation of WAD2M wetland detection in the Australian desert, which is mitigated by the small $C_{substrate}$ over this region when $C_{substrate}$ is considered instead of the uniform substrate (Fig.5).

The ensemble mean of the GMB LSMs simulations (Saunois et al., 2020) is presented Fig.6.e. Detailed maps of the individual model outputs are provided in the Supplementary Fig.S3, together with the LSMs output standard deviation map. Comparison is made with GMB LSMs run in diagnostic mode, i.e. all LSMs were run with the same wetland area WAD2M

standardized to the same 1°x1° grid for consistency. In addition, Fig.6.f shows the model mean of the WetCHARTs ensemble (Bloom et al., 2017), which incorporates different wetland extent products. In the WetCHARTs ensemble, three scaling factors are tested to amount to a global mean annual flux of 124.5, 166 or 207.5 TgCH4 yr$^{-1}$ (Saunois et al. (2016) lower, mean and upper estimates). Here we have only selected the members of the ensemble that were calibrated to the mean budget (166 TgCH4 yr$^{-1}$). The standard deviation map of methane emissions from the WetCHARTs ensemble is also included in Supplementary

Fig.S3.

The spatial distribution of SatWetCH4 emissions run with WAD2M is similar to the average of the LSMs ensemble run with the same wetland extent over America, Australia, and Europe. However, there is considerable variability in the spatial emissions between models in some regions, including the Siberian lowlands (Ob), Australia, India, and over sub-equatorial Africa, even though the same water surface map is prescribed.

In subequatorial Africa, emissions are highly uncertain from one model to another. The different GMB model outputs show a wide range of emissions (Supplementary Fig.S3). Four of the LSMs have low emissions (<0.1 gCH4/m$^2$/month), while the other nine have moderate to high emissions (0.1 to 0.5 gCH4/m$^2$/month). Like the first group of LSMs, the WetCHARTs ensemble mean and the SatWetCH4 model predict almost negligible emissions (<0.05 gCH4/m$^2$/month), while the LSM ensemble mean estimates emissions around 0.1 gCH4/m$^2$/month. The number of measurements available to evaluate the simulations is

limited (difficult to access areas, no flux towers, no in situ flux or concentration measurements). A hypothetical underestimation of substrate availability $C_{substrate}$ in this region could be attributed to cloud cover limiting visible and near-IR observations. Indeed, since the PsnNet parameter of the MODIS parameter is low in this zone, the $C_{substrate}$ dataset estimates a very low available substrate.



Most model outputs show Australia with low emissions. However, some models produce surprising spatial patterns over
Australia, especially in desert regions for LPJ-GUESS and TEM-MDM. This is due to a problem in WAD2M, as discussed in
3.3.2, which presents a non-zero fraction of wetlands in this Australian desert region. However, other models, including ours,
certainly mitigate this issue by reducing emissions due to other parameters such as vegetation cover or hydrological settings,
thereby compensating for the problem of misclassification of wetlands.

Northern India also exhibits lower emissions in SatWetCH4 when run with WAD2M compared to the GMB average. Sup-
plementary Fig.S3 indicates that this elevated average is mainly due to one model, DLEM, with very high emissions in this
region, while the other models show emissions similar to ours. This discrepancy raises questions about the representation of
rice paddies in the DLEM model despite the forcing of water surface dynamics.

Overall, the spatial distribution of SatWetCH4 run with WAD2M aligns with the ensemble of LSMs and their uncertain-
ties. We have discussed that when SatWetCH4 is run with TOPMODEL, different spatial patterns emerge, which are no less
surprising when compared to the variations observed in the GMB LSMs and WetCHARTs simulations.





### 3.3.4   Latitudinal and seasonal variation of methane emissions

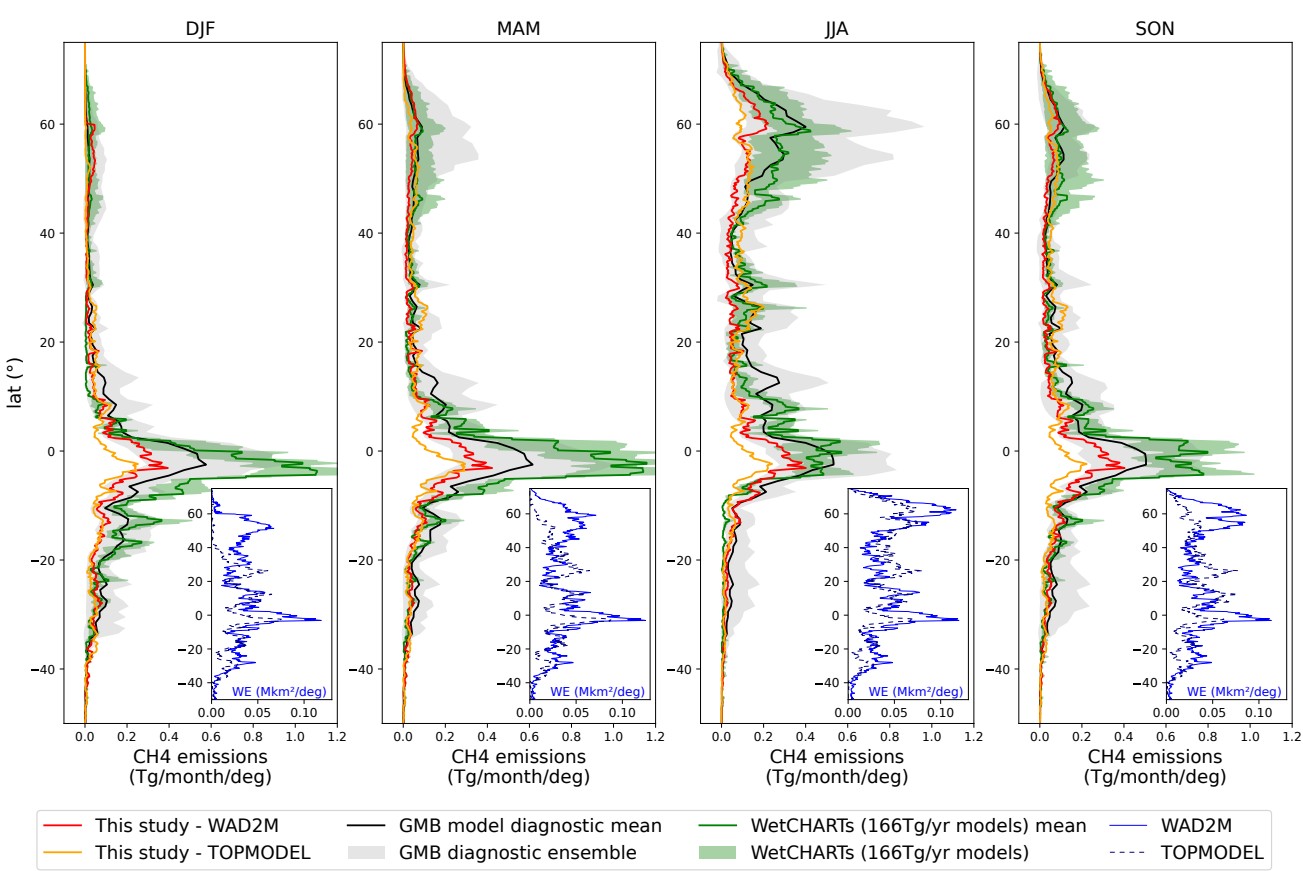

**Figure 7.** Latitudinal distribution depending on the season of wetland methane emissions from SatWetCH4 run with WAD2M (red) or TOPMODEL (orange), from LSMs (filled grey) with LSMs average (black), and from WetCHARTS models calibrated with 166 Tg CH$_4$/yr budget (filled green) with ensemble average (green). WAD2M and TOPMODEL wetland extents seasonal mean are also presented in inserts (blue resp. solid and dashed lines). LSMs estimates are those contributing to the GMB (Saunois et al., 2020), all run with the same wetland extent product (WAD2M). All representations are 2003–2020 seasonal means.



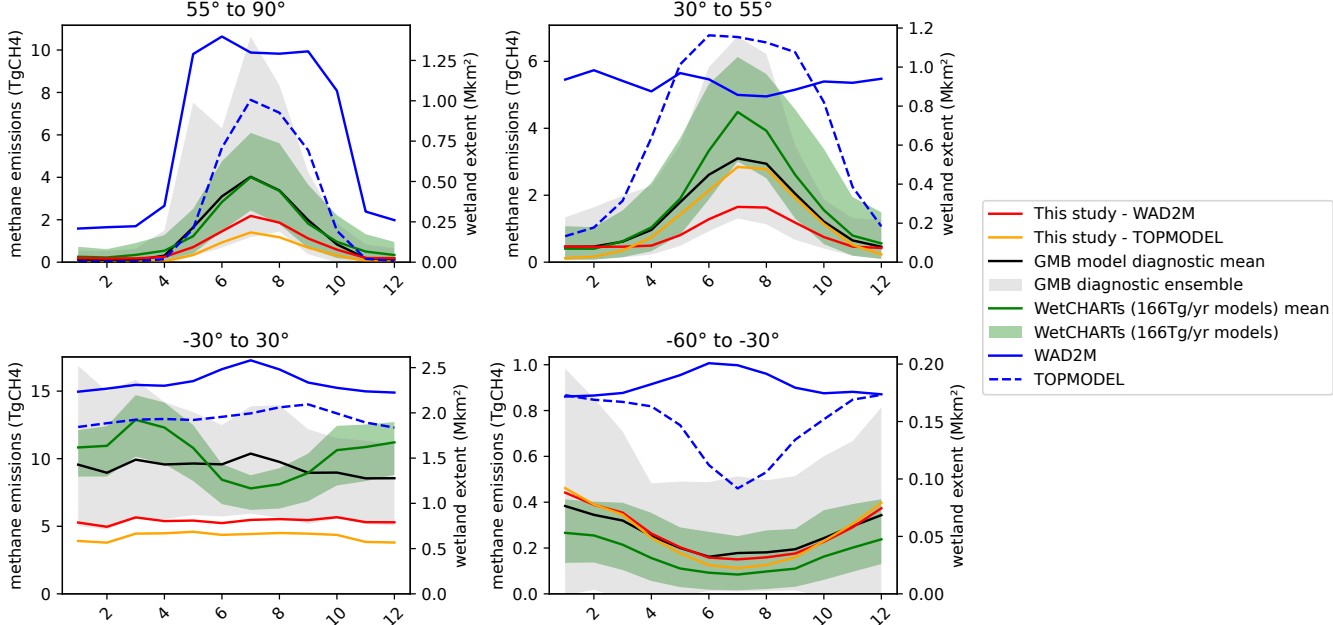

**Figure 8.** 2003-2020 CH$_4$ emission mean per month per latitudinal band from SatWetCH4 run with WAD2M (red) or TOPMODEL (orange), from LSMs (filled grey) with LSMs average (black), and from WetCHARTS models calibrated with 166 Tg CH$_4$/yr budget (filled green) with ensemble average (green). WAD2M and TOPMODEL monthly wetland extent 2003-2020 means are presented in blue. LSMs estimates are those contributing to the GMB (Saunois et al., 2020), all run with the same wetland extent product (WAD2M).

Fig. 7 shows the latitudinal distribution per season for SatWetCH4 run with WAD2M and TOPMODEL, as well as the GMB LSMs and WetCHARTs ensemble. The monthly variation for emissions and wetland extent per latitudinal band is shown in Fig. 8. Note that the WetCHARTs models are calibrated to the GMB annual budget and are therefore not independent in terms of methane emission amplitude. SatWetCH4 is in the lower range of the GMB LSMs (grey areas), or even slightly

below this range in the 30°S-30°N band. The total annual budget of SatWetCH4 wetland emission estimate averages 85.6 Tg CH$_4$ yr$^{-1}$ with WAD2M (resp. 70.3 with TOPMODEL), which is lower than the range of the GMB LSMs estimates (102 to 181 Tg CH$_4$ yr$^{-1}$). This discrepancy is consistent with the underestimation of methane fluxes by SatWetCH4 at tropical sites (discussed in section 3.2). Note that this difference in total emissions could be easily resolved by calibrating the *k* parameter to

the total emissions of the mean GMB LSMs if we need to constrain total emissions, as it has been done previously by Bloom et al. (2017); Gedney et al. (2019).

We note that SatWetCH4 simulation with TOPMODEL estimates lower emissions in the tropical and boreal bands compared to the simulation with WAD2M (Fig. 7). This is consistent with the smallest wetland extent of TOPMODEL over these regions, as non-inundated peatlands are not considered in TOPMODEL. Note also the higher fluxes obtained in the simulation with

TOPMODEL than with WAD2M around 25-30°N, due to the larger wetland extent of TOPMODEL over Asia. The latitudinal



distribution of SatWetCH4 (Fig. 7) is consistent with the distribution of the LSMs ensemble, except for the African subequatorial band mentioned earlier. SatWetCH4 reproduces similar seasonal changes as the GMB LSMs (Fig. 7), while the latitudinal distribution of the WetCHARTs ensemble presents larger emissions in the 15°S-5°N band in the DJF, MAM and SON seasons (mainly due to high emissions in the Congo region, visible in Fig. 6).

This different seasonal cycle in the tropical band (30°S-30°N) for WetWHARTs ensemble is also visible on Fig. 8, while there is an absence of a pronounced seasonal pattern, both in terms of emissions and in terms of wetland extent for our model and the GMB models. This difference in tropical seasonal cycle could be due to the wetland extent used in WetCHARTs. For the boreal region (55-90°N), we find that the seasonal variation of the simulated emissions from our model is close to that of most GMB LSMs, as it is in the northern temperate band (30-55°N). However, the wetland extents of WAD2M and

TOPMODEL show very different seasonality, particularly in the northern temperate band (30-55°N), where WAD2M has a more stable wetland extent than TOPMODEL. Indeed, the methane emission seasonality in the boreal and temperate regions is mainly driven by temperature, which explains these similar seasonal cycles in emissions, although the seasonal cycles in wetland extent are different. For the southern temperate band (60°S-30°S), WAD2M and TOPMODEL exhibit contrasting seasonality in wetland extent, but the simulated seasonal variations in emissions are close because, as expected, temperature

drives the variability of methane fluxes in this temperate region.



### 3.3.5 Interannual variability in methane emissions at basin scale

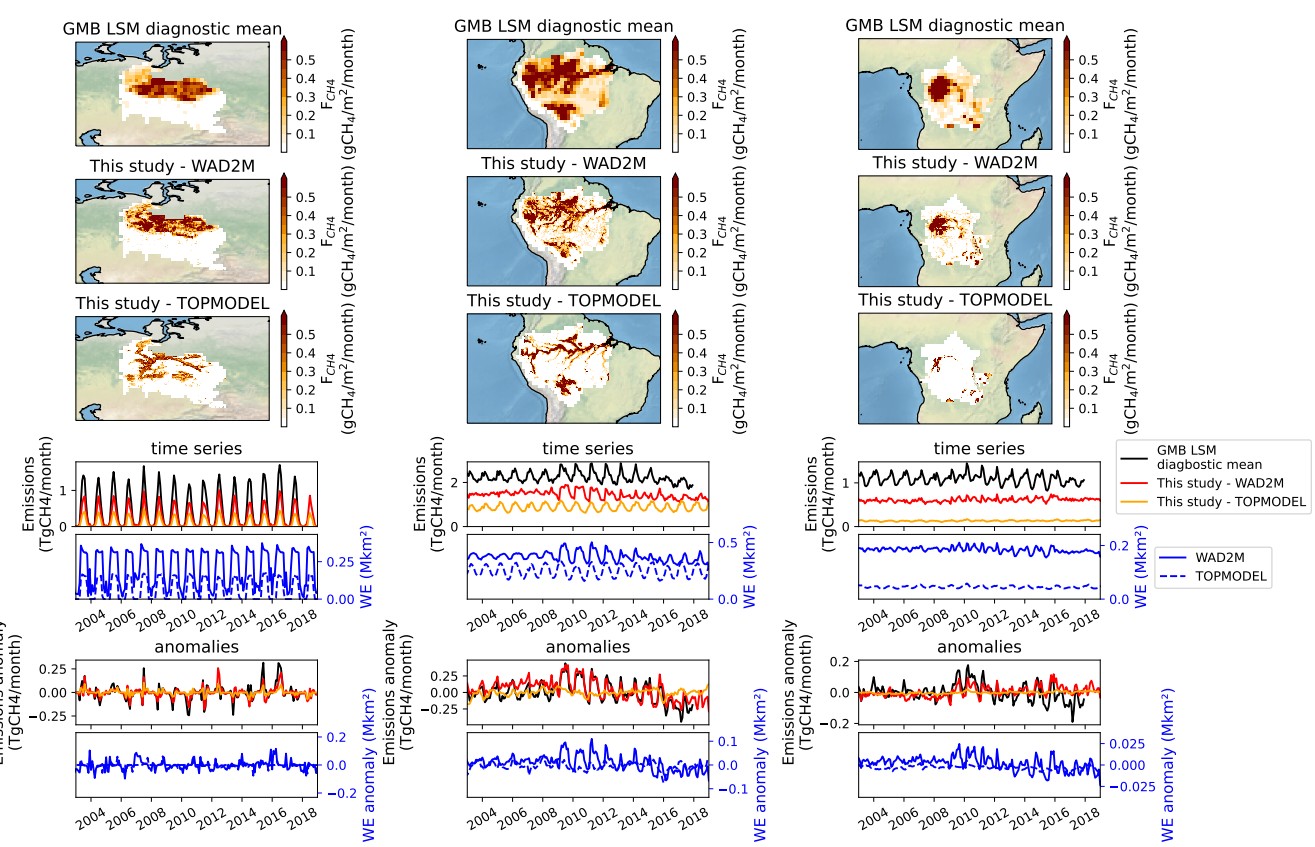

**Figure 9.** Methane emissions for different basins: the Ob, the Amazon, and the Congo. The maps show the spatial pattern of methane emissions from the GMB LSM mean (top map) and SatWetCH4 simulations with WAD2M (middle map) or TOPMODEL (bottom map). The lower panels represent the sum of methane emission time series and deseasonalized anomalies over the basins of SatWetCH4 simulations with WAD2M (red) or TOPMODEL (orange), GMB LSMs (dashed grey), and LSM mean (black). All LSMs were run with the same WAD2M wetland extent.

Fig. 9 depicts SatWetCH4 model and GMB LSMs emissions simulated with WAD2M and their anomalies for three basins: the Amazon, the Ob, and the Congo. Also shown are wetland areas and their anomalies over these basins.

In the Amazon and Congo basins, notable amplitude irregularities were observed when using WAD2M. Two regime changes are observed in the WAD2M extent around 2009 and 2014, probably due to inter-calibration problems caused by satellite changes in the original SWAMPS surface water product. Surprisingly, the average of the LSMs is less affected, even though the LSMs are forced with the same water surface. However, on closer examination of individual LSMs (see Supplementary Fig.S4), we see that some LSMs are as affected as SatWetCH4 by these inconsistent water surface changes, while others are




less affected. We deduce that these models, which are not affected by the WAD2M temporal changes, must have parameters

that interfere with the consideration of the wetland surface. TOPMODEL suggests more consistent time series in terms of wetland extent (softer variations), which also allows for more realistic variations in terms of emissions.

## 4   Model limitations and outlook

The simplified approach used here as a one-step model allows for some quick and easy simulations, representing major first order phenomena affecting methane emissions from wetlands. While presenting a smaller annual budget, due to a possible

underestimation of the magnitude of emissions, we found that this formulation presents realistic spatial and temporal variations when compared to other more complex and computationally intensive models. By scaling the $k$ factor to a target estimate, the discrepancy in global emissions could be easily resolved.

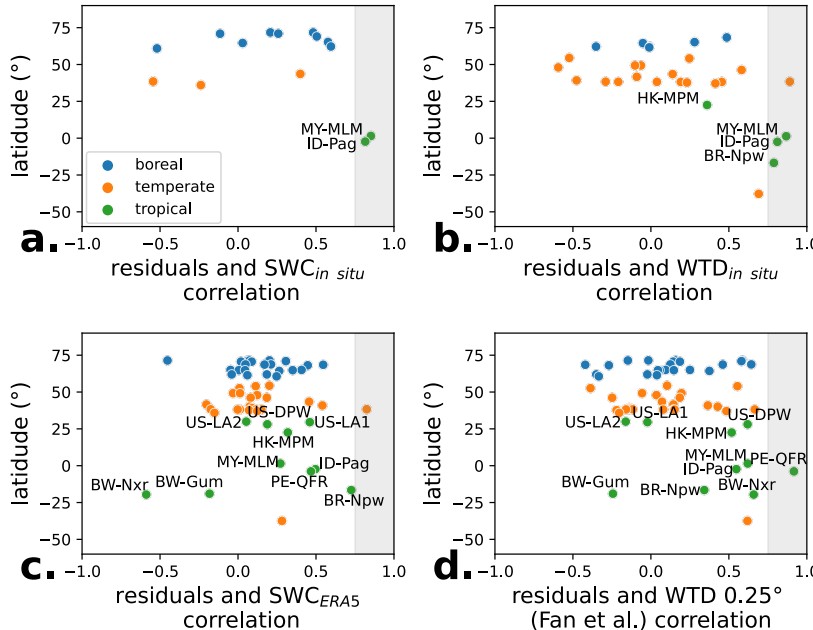

**Figure 10.** Correlation of residuals (observation-predictions) with **a.** in situ WTD, **b.** in situ SWC, **c.** 0.25° ERA5 SWC and **d.** 0.25° WTD (Fan et al., 2013). These residuals are calculated for a single site calibration of SatWetCH4 in order to remove the seasonal cycle that the model can capture through its variables (soil temperature and substrate availability). Grey background represents r>0.75.

Some refinements could be considered to improve the accuracy of the model. We found that the simulated temporal variability is less well captured at tropical sites than at temperate and boreal sites, as temperature does not drive seasonality in

these regions. In fact, some studies (Kuhn et al., 2021; Knox et al., 2021) suggest that methane emissions in tropical regions are influenced by WTD. To investigate the flux dependence on a local water parameter, we calculated residuals from the single site calibration presented in section 3.2. A residual is the difference between observed and predicted methane fluxes, and thus



represents the error of the model at a given site at a given time. Fig. 10 illustrates the correlation of different hydrological variables with the residuals. In the tropics, the missing variability appears to be strongly linked to soil water variations: 2 out

of 2 (MY-MLM and ID-Pag) tropical sites monitoring SWC show a strong temporal correlation (r > 0.75) of residuals with locally measured SWC, and 3 out of 4 sites monitoring WTD (HK-MPM, MY-MLM, ID-Pag and BR-Npw) show a strong temporal correlation (r > 0.75) of residuals with locally measured WTD.

To test whether this site-level correlation could be used in SatWetCH4 model, we repeat this experiment using global datasets at 0.25° of SWC and WTD. For each site, we selected the nearest 0.25° pixel of the ERA5-Land monthly averaged SWC dataset

(available at https://cds.climate.copernicus.eu/)), and the nearest pixel of the WTD from Fan et al. (2013) aggregated at 0.25° (as only one typical year is provided, this year is replicated for all years of the in situ flux measurement period). Fig. 10.c and d. show the resulting correlation of these two variables at 0.25° with the residuals. None of the 11 tropical sites show an r > 0.75 between residuals and ERA5 SWC and only one site (PE-QR) shows an r > 0.75 between residuals and 0.25° WTD. This is due to the fact that these 0.25° datasets poorly represent the temporal variations measured in situ, as shown in Supplementary

Fig.S2 for the ERA5 SWC. SWC and WTD in wetlands have very spatially localized specificities and variations. Furthermore, the small number of sites available in the tropics (11) makes it even more difficult to find an empirical relationship with a water variable. We were unable to include this important parameter at SatWetCH4 model resolution of 0.25°. The 100m satellite-derived SWC obtained by Planet (De Jeu et al., 2014) could be examined and the model run at finer resolution. In fact, Albuhaisi et al. (2023) found an improvement in their model for the boreal region when using this high resolution product.

Further research could be conducted to see if similar results are obtained in the tropics, where this parameter is most needed. Unfortunately, this product is not freely available.

It is worth noting that the site level comparison of modeled fluxes with observations assumes that the sites are all wetlands ($f_w = 1$), without any temporal variation. However, when the SatWetCH4 model is run, this wetland fraction is dynamic, introducing seasonality due to water and partially compensating for the lack of a local water parameter.

Another limitation is that the consistency of the time series of methane emission estimates at the catchment scale is strongly affected by errors in the WAD2M database. This makes it difficult to study interannual variability or trends. The TOPMODEL time evolution does not have these major temporal inconsistencies, but it is based on a hydrological model and not on satellite observations. It also does not include non-inundated peatlands. An improved satellite-derived dynamic wetland surface map would be crucial to address these issues while maintaining observational data in our data-driven approach.

The simplified SatWetCH4 model we have developed makes important approximations that imply important shortcuts. In particular, no distinction is made between methane production and emissions. This supposes that SatWetCH4 one-step equation includes production, oxidation, and transport in a single formulation, which are sometimes distinguished in some of the more complex LSMs (Wania et al., 2013; Morel et al., 2019; Salmon et al., 2022). Among the 3 pathways of methane transport in wetlands, including diffusion, ebullition and plant-mediated transport, plant-mediated transport is the dominant one (Ge

et al., 2024). Ge et al. (2024) have recently published a comprehensive review of the role of plants in methane fluxes, showing their influence not only on methane transport but also on methane production and oxidation. Feron et al. (2024) also show




that trends in methane flux changes at the site level depend on ecosystem and vegetation type. Accounting for the different vegetation classes therefore appears to be a possible improvement to our simplified approach.

A simple way to account for this in SatWetCH4 model at a first order would be to fit the scaling factor $k$ and/or $Q_{10}^0$ as a function of vegetation class or wetland type. We performed such calibration tests, taking into account the wetland classification. However, the cost function either did not converge due to the small number of sites per category, or the result was highly dependent on a few sites. In fact, eddy covariance flux towers measuring methane emissions are not evenly distributed around the globe and their distribution is highly skewed, as discussed in part 2.2. Some wetland categories are poorly represented, for example, there are only two mangrove sites. This scarcity of data makes this type of calibration highly uncertain. However, we can expect an improvement in the coming years, as in situ methane measurement is a rapidly growing field, as shown by the increasing number of flux towers along the years in the Supplementary Table S1. Future data, especially in the tropics, will be essential to better constrain the models and to include more processes into account.

## 5 Conclusions

SatWetCH4 model was developed to simulate global wetland methane emissions at 0.25°x0.25° with a monthly time resolution. This data-driven approach was calibrated with 58 sites of eddy covariance flux data, allowing an approach independent of other estimates. Most of SatWetCH4 model input variables are derived from satellite observational products. In particular, a new estimate of the substrate availability was derived using MODIS-derived NPP. This product, called $C_{substrate}$, appears to be a more realistic approach than previous studies that considered all SOC as available carbon.

At the site level, the SatWetCH4 model reproduces well the boreal fluxes and most of the temperate fluxes, but poorly the emissions seasonality of the tropical sites. This could possibly be improved in future studies by adding high resolution information on local water availability (SWC). Another important improvement would be a calibration per wetland type, which would allow the influence of vegetation to be taken into account as major transport pathways. For this, more eddy covariance flux measurements in the tropics are essential to gain a deeper insight into the processes governing temporal variations in this latitudinal band, and to develop and calibrate this one-step model.

This simple formulation, allows a quick (within a few seconds) simulations over decades. Compared to the GMB LSMs, SatWetCH4 model shows consistent spatial patterns and seasonal variations. However, it is below the range of the LSMs in terms of budget: 86 Tg $CH_4$ yr$^{-1}$ (estimated using WAD2M) and 70 Tg $CH_4$ yr$^{-1}$ (estimated using TOPMODEL), while the LSMs present a global range of 102-182 Tg $CH_4$ yr$^{-1}$. This underestimation is partly due to the scarcity of eddy covariance data in the tropics, leading to an underrepresentation of high emitting tropical sites.

Finally, we found some inconsistencies in the widely used WAD2M surface wetland extent. A new wetland map is currently being produced (Bernard et al., in prep.), based on GIEMS-2 (Prigent et al., 2020) observations, which provide a seamless estimate of inundated areas with realistic interannual variability (Bernard et al., 2024). Applying SatWetCH4 model with this new dataset would allow the study of annual variability and trends in emissions.



Another perspective is the coupling of SatWetCH4 with atmospheric inversions. Indeed, one way to overcome the challenges

associated with calibration using surface flux data is to incorporate this simple model into an atmospheric inversion model. This would allow the optimization of both parameters $k$ and $Q_{10_0}$ in the inversion equation using atmospheric concentrations (more numerous than methane fluxes data, especially with satellite data), rather than just the optimization of the methane flux value, as is usually done in inversion models.

*Code availability.*    The optimization and model codes are available at https://doi.org/10.5281/zenodo.11204999.

*Data availability.*    All global forcings used in this study are freely available online: ERA5-Land data at https://cds.climate.copernicus.eu/, WAD2M at https://zenodo.org/records/3998454, TOPMODEL derived wetland extent at https://zenodo.org/records/4571667, and MODIS PsnNet data through https://lpdaac.usgs.gov/products/mod17a2hgfv061/.

*Author contributions.*    J.B., M.S., E.S., P.C., S.P. and A.B. conceived the main ideas of this study and contributed to the research. J.B. built the model, ran the simulations and performed the numerical analyses. J.B. drafted the manuscript with input from M.S. and E.S.. J.B, M.S,

E.S, P.C., S.P., P.S-O. provided critical feedback on the manuscript. P.G., P.S-C. and J.J. each provided data from a flux tower.

*Competing interests.*    Authors declare no competing interests.

*Acknowledgements.*    J.B. is funded by a PhD grant from the Institut National des Sciences de l'Univers (INSU) of the Centre National de la Recherche Scientifique (CNRS). J.B. would like to thank Vladislav Bastrikov for his insightful help in designing the model calibration. The authors would like to thank the managers of the 55 out of 58 eddy covariance flux towers who made their data available as open source. These

data play a crucial role in improving our understanding of methane emissions from wetlands and in calibrating the models more accurately.



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
