# Peer review of "Satellite-based modeling of wetland methane emissions on a global scale (SatWetCH4 1.0)"

_EGUsphere, 2024_

## Author Comment (AC1)

**Replies to Referee 1**

The article titled "Satellite-based modeling of wetland methane emissions on a global scale (SatWetCH4 1.0)" presents a pioneering approach to understanding and quantifying methane emissions from wetlands across the globe. The authors have developed an innovative model that leverages satellite-based data to simulate methane emissions with a high degree of spatial and temporal resolution. This study is particularly significant given the substantial contribution of wetlands to global methane emissions and the critical role of methane as a greenhouse gas.

The manuscript is well-structured, with a clear abstract that succinctly summarizes the study's objectives, methods, and findings. Overall, this study represents a significant advancement in the field of wetland methane emissions modeling. It is my pleasure to recommend this paper for publication in GMD journal.

Thank you for your positive feedback on the manuscript. We are honored to read that you find our approach innovative and our model important for understanding and quantifying global wetland methane emissions.

My comments here are relatively minor, but I hope may be useful for the authors to consider.

We greatly appreciate your time and effort in reviewing our work. Thank you for your careful reading of the paper and for raising interesting methodological considerations. Hereafter are point-by-point responses to your comments. We hope that we addressed all of your questions and suggestions.

The line numbers given hereafter refer to the tracking-changes PDF manuscript. Changes in the manuscript are given here in violet.

1. L165-L171: Is the reason for using ERA5's lay2 soil moisture because of the high correlation coefficient between the site data and lay2 soil moisture? In this article (DOI: 10.1126/sciadv.aba2724), the soil moisture used is from 0-289cm depth, and I haven't seen how the interannual variation of soil moisture at other layers correlates with the site observations. As far as I know, the interannual variation of soil moisture in lay1 and lay2 is similar, and lay1 is closer to the surface soil moisture. Why not use the data from lay1?

The soil ERA-5 reanalysis data contains 4 layers : layer 1: 0 - 7 cm, Layer 2: 7 - 28cm, Layer 3: 28 - 100cm, Layer 4: 100 - 289cm. We used in this paper the ERA 5 data for two variables (soil temperature-described lines 166 - 172, and the soil moisture - SWC). We used SWC for tests only (Sect 4, line 390-400 of the preprint). Since we are unsure if your comment is on soil temperature or soil moisture, we address both hereafter.

In the model, we use ERA-5 soil temperature. In the literature, methanogenesis is expected to be the highest in the upper saturated layers (Bridgham 2013, Nzotungicimpaye 2021), with maybe highest methanogenesis rates in the ~30 upper cm (Cadillo-Quiroz 2006, Valentine 1994, Priemé 1993). We chose *layer 2* (7 - 28cm) for ERA 5 product to be more representative for the layer where methanogenesis is expected to occur, as the top ERA5 layer is only 7 cm deep. We checked ERA 5 soil temperature, and indeed the temperature of layer 1 and layer 2 were highly correlated ($R^2 > 0.99$). Layer 2 was also slightly better correlated with site measurements than layer 1 (certainly partly due to probes depth). The difference between layer 1 and layer 2 is very small, and no changes in model output were finally observed using soil temperature layer 1 instead of layer 2.

We later discussed the influence of Soil Water Content (SWC) using the data from ERA 5 layer 2 to be consistent with the temperature. Following your suggestion, we checked the differences between SWC layer 1 and SWC layer 2. We repeated the numerical experiments with SWC layer 1 and the conclusions remain the same as with layer 2 (see manuscript L388-395 and Supplementary L19-23). We join here the corresponding Figures. In Fig. 10 we see that the time correlation of residuals with SWC from ERA 5 is below 0.75 for almost all sites, both with layer 1 and layer 2. In adapted Fig.S2 we can see that the SWC layer 2 doesn't reproduce well local wetland SWC measured in situ, especially temporal variations.

[Figure]

Article Fig.10 adapted : residuals correlation with a. SWC in situ, b. SWC lay2, and c.SWC lay 1

[Figure]

Supplementary Fig.S2 with SWC layer 2 (original preprint supplementary Figure)

[Figure]

Supplementary Fig.S2 adapted with SWC layer 1

2. L206-L210: Are you suggesting that the green single-peaked line (Albuhaisi et al., (2023)) is more in line with physical laws but the formula is more complex and prone to non-convergence, which is why such a complex approach was not adopted? So, your point is that the methods of Albuhaisi et al., (2023) should be more reasonable, rather than the others?

[Figure]

**Figure 2. a.** Comparison of $Q_{10}(T)$ formulation with Walter and Heimann (2000), Gedney (2004), WetCHARTs Bloom et al. (2017), WETMETH Nzotungicimpaye et al. (2021), and Albuhaisi et al. (2023). **b.** Effect of the different $Q_{10}(T)$ formulations when incorporated in the temperature dependency function.

We believe that you are asking whether the WETMETH approach (the green line) (Nzotungicimpaye et al. (2021)) should be more reasonable. In fact, Nzotungicimpaye et al. (2021) proposed a formulation based on Dunfield et al. (1993); Metje and Frenzel (2007); and Schipper et al. (2014). This formulation enables a temperature optimal range for methanogenesis around 25°C-30°C and include a Q10 formulation as:

$$Q_{10}(T) = 1.7 + 2.5\ tanh(0.1(T_{ref} - T)) \quad \text{with} \quad T_{ref} = 308.15\ °K$$

In their equation, the Q10(T) formulation is set using common Q10 literature values order of magnitude, however this formulation was not fitted with in situ fluxes. During our model development, we derived a Q10(T) equation to have a similar shape using a formulation depending on temperature, having an optimal 25°C-30°C temperature range for methanogenesis, and for which we could fit a $Q_{10\ 0}$ parameter like in Gedney formulation.

However, when optimizing the parameters with in-situ fluxes, this version of the model did not converge as well as with the simple formulation and, as you mentioned, made the equation more complex. The modeled methane fluxes (local and global) were similar using a simpler formulation. For these reasons, we preferred keeping the simple Gedney version in the model.

Albuhaisi et al., (2023) formulation, on the contrary, has less physical meaning than Nzotungicimpaye et al. (2021). Their formulation produces some abrupt changes, especially an abrupt drop of methanogenesis rates at T = 30°C (Fig. 2.b) which makes it non-relevant for global studies as some tropical wetlands can experience temperatures around 30°C.

The text was modified (lines 210-218 of manuscript with tracking changes) to explain this more clearly, also with respect to comment 3.

3. Fig 2: By the way, could you explain why there are three lines in WetCHARTs, and what do they represent, respectively? What is the significance of such classification? Your results seem to be mainly close to the one with Q10=3.

In WetCHARTs, Bloom et al. used different datasets and parameter values, including 3 different Q10 values (1, 2, and 3). The WetCHARTs ensemble was created by using 3 Q10 values × 9 heterotrophic respiration estimations × 4 wetland extent maps × 3 scaling factors = 324 estimates. This was done to assess the sensitivity of these constraints to the role of carbon, water, and temperature variability in the global spatial and temporal variability of wetland CH4 emissions. We have modified the text of section 3.1 to explicitly explain the different Q10 values shown in Figure 2. The Q10 in WetCHARTs is of a different nature, as the WetCHARTs equation uses heterotrophic respiration, which already contains a Q10. The Q10 from WetCHARTs was then removed from the comparison, as described in the discussion with the Referee #3 (comment 2.).

4. L221: I understand that when you were assessing with the site data, you set $f_w$ to 1, and later when generating the product, you utilized WAD2W and TOPMODEL? Or after considering the two different wetland areas, why didn't you compare with this quantity? Was it because you considered the spatial resolution of the gridded data to be too coarse?

For site calibration, we expect the flux tower footprint to be completely - or mostly - covered by wetlands. Therefore, we fit the equation with fw = 1 as you mentioned. Eddy covariance tower footprints are typically between 100 m² and 10,000 m² (BALDOCCHI et al., 2001; KUMAR et al., 2017).
We considered this approximation to be more accurate than using the WAD2M and TOPMODEL fractions at local sites, as WAD2M and TOPMODEL resolution are quite coarse (0.25°x0.25° pixels ~ 600 km² ~ 600 000 000 m²) and represent the wetland fraction over larger areas than the wetland sites.

5. In Figure 3c, the simulation appears to be quite similar to the observations in the low-value area, but there is a significant difference in the high-value area. Could you briefly explain the reason for this? Is it related to the wetland area $f_w$ being set to 1?

In this graph, SatWetCH4 with the same optimized parameters is applied to all sites. We think the main reason for having the highest differences at the highest flux values is that we did a regression to optimize the fluxes. When minimizing the mean error, a regression tends to flatten the extreme values in estimates. This leads to overestimating some sites (mainly those with lower FCH4 fluxes) and underestimating others (mainly the highest fluxes). Salmon et al. 2022 (https://doi.org/10.5194/gmd-15-2813-2022) have shown these trends in their figure 7, where the multisite optimization is similar to the optimization done here. Moreover, the sites with high mean flux values (mainly tropical sites) are little represented in the flux tower distribution ; and thus have smaller weight in the cost function.

For some sites with high fluxes (tropical and temperate sites), errors in the temporal variations could be partly explained by the lack of a variable for local water, such as soil water content or water table depth (figure 10.a and 10.b). Such variables, if we were able to take it into account, would introduce a different spatial pattern between sites, and hopefully increase the model accuracy in areas with the largest errors.

6. L227-L230: From a statistical relationship perspective, it is true that the correlation coefficient between the tropical region and the site data is low. Could you briefly explain the reason? Since your model uses MODIS observations similar to NPP, there should theoretically be more data in the tropical regions. Is it because the emissions are high in the tropics and the mechanisms of methane production activities are not well understood?

The authors believe that indeed, the mechanisms regulating methane production in the tropics is not well represented in this simple model. The model variability is driven only through the wetland extent dynamics and the temperature. However, temperature is not the main driver for the dynamic of wetland emission in the Tropics.

We discuss in Section 4 the role of a local water parameter. We noticed that the model residuals (i.e., errors defined by observation-prediction) of tropical sites and some temperate sites are correlated with a local water parameter. Thus the water table depth or the soil water content could partly explain the temporal variations missed by the model. SWC and WTD are known to be drivers of the seasonal variations over the places where temperatures have small seasonal variations, such as the Tropics. We added a sentence in section 3.2 referring to section 4 : *"Furthermore, the mechanisms driving the temporal variations in tropical methane flux are certainly poorly represented in the model, as discussed in Sect.4."*

MODIS NPP observations are used to assess substrate availability. The temporal variations of $C_{substrate}$ are rather small (see Figures below). The simulated fluxes reflect on the (spatially heterogeneous) substrate available, weighted (in time and space) by other variables such as the temperature.

[Figure]

7.  In Figure 4, should the time period range of the annual average be indicated in the figure title? Is it 2003-2020? In line 249, it is mentioned that WAD2M is from 2003 to 2020, is TOPMODEL climatological? Please clarify.

Thank you for your comment. WAD2M is available for 2000-2020 and TOPMODEL 1980-2020, both with monthly resolution. Here we have indeed made a 2003-2020 mean to be comparable with the model run period 2003-2020 (as the time period is limited by PsnNet MODIS data). We have added this detailed information about the considered period in the caption of Figure 4.

8. Line 261: The absolute magnitude of the $C_{substrate}$ value is small, and the text provides the corresponding explanation. My question is, after normalization, compared to the other two products (SoilGrids and HWSD), the high-value areas of $C_{substrate}$ in the region north of 60°N and more northerly, could you briefly explain this reason?

These high amount of carbon are due to the turnover rate K(T) value in Eq. 2 that depends on temperature:

$$K\,(T)\;=\;K^{ref}\,Q_{10K}^{(T-TKref)/10}$$

Some easily available carbon is created in summer through NPP in these regions north of 60°N, and not degraded by heterotrophic respiration during winter due to lower turnover rates K(T) because of the low boreal temperatures.

We had run some tests with a fixed turnover rate (K = 0.5 yr⁻¹); i.e, without taking into account the temperature dependency for the heterotrophic respiration. This results in lower carbon amounts (<2kg/m²) in the northern regions, as depicted in the following Figure (left column).

Figure 5 of our prepint manuscript shows that the $C_{substrate}$ spatial distribution is smoother than in SoilGrids and HWSD which are representing total Soil Organic Carbon (SOC). Indeed, SOC and $C_{substrate}$ proxy do not represent the same variable : SOC is mainly contained in peatlands and other organic rich soils, which is not the case for our $C_{substrate}$ proxy that represent easily available organic carbon and is then .

[Figure]

9. L293-294: The overall smaller values of $C_{substrate}$ are due to the lower values obtained from MODIS. Do you think these comparatively smaller values are reasonable? Please explain.

As we fit the methane emissions using a scaling parameter, simulation outputs are fitted so that the equation - that comprises $C_{substrate}$ variable - fits the fluxes. Then, the order of magnitude of $C_{substrate}$ doesn't alter the fluxes order of magnitude as the scaling factor k is adapted.

L293-294 are referring to the fact that $C_{substrate}$ is small (<0.5kgC/m²) over subequatorial Africa due to small values of the MODIS PsnNet input (Fig.5). This leads to small fluxes (<0.05gCH4/m2/month) in SatWetCH4 with both WAD2M and TOPMODEL (Fig.6.a and b.). We discuss this in L310- 318 of the revised version:

*"In subequatorial Africa, emissions are highly uncertain from one model to another. The different GMB model outputs show a wide range of emissions (Supplementary Fig.S3). Four of the LSMs have low emissions (<0.1 gCH4/m2 /month), while the other nine have moderate to high emissions (0.1 to 0.5 gCH4/m2 /month). Like the first group of LSMs, the WetCHARTs ensemble mean and the SatWetCH4 model predict almost negligible emissions (<0.05 gCH4/m2 /month), while the LSM ensemble mean estimates emissions around 0.1 gCH4/m2 /month. The number of measurements available to evaluate the simulations is limited (difficult to access areas, no flux towers, no in situ flux or concentration measurements). A hypothetical underestimation of substrate availability $C_{substrate}$ in this region could be attributed to cloud cover limiting visible and near-IR observations. Indeed, since the PsnNet parameter of the MODIS parameter is low in this zone, the $C_{substrate}$ dataset estimates a very low available substrate."*

The authors think that the SatWetCH4 emissions could indeed be underestimated over this region due to MODIS data which constrains $C_{substrate}$ low values. The problem could also arise from an overestimation by other models due to uncertain water detection there in WAD2M. But the lack of flux data over this area makes it hard to know which models are right.

10. L356-358: Why did you previously mention that the global distribution is related to the distribution of wetland areas, yet the wetland distributions in the Southern Hemisphere are opposite for the two, yet the methane emission fluxes from wetlands are similar? In Equation 1, temperature and wetland area are directly proportional to methane emission fluxes; why is temperature considered the dominant parameter in the Southern Hemisphere?

Indeed, simulated CH4 fluxes spatial distribution depends on the wetland distribution, the substrate pattern, and the temperature. Each of these variables have different contributions in terms of spatial pattern and temporal variation.

As the temperature dependency is exponential, the temperature is expected to dominate the seasonal cycle in boreal, and temperate regions (both northern and southern temperate regions) where the temperature variations are important. As a result, the shape of methane emissions seasonality observed in the temperate Southern Hemisphere is driven by the temperature seasonality (high in Nov-March during southern summer).

---

## Author Comment (AC2)

**Replies to Referee 2**

**Review of Satellite-based modeling of wetland methane emissions on a global scale (SatWetCH4 1.0). Bernard et al.**

The authors use surface eddy covariance data for wetland methane fluxes from the global FLUXNET-CH4 Community Product v1.0 to calibrate an empirical model that is applied to produce a globally upscaled wetland methane emissions product. The study is at the forefront of global wetland methane modeling and responds to the scientific imperative to understand the global methane budget in a time of rapid change (accelerating growth of atmospheric methane concentrations since 2014). With revisions focused on reconciling findings of the author's study (SatWetCH4 1.0) with the first and only published global wetland methane upscaling product (UpCH4 v1.0; McNicol et al. 2023) and providing more justification of the model calibration approach, the study is likely to become an important contribution to wetland methane upscaling science.

*McNicol, G., Fluet-Chouinard, E., Ouyang, Z., Knox, S., Zhang, Z., Aalto, T., et al. (2023). Upscaling wetland methane emissions from the FLUXNET‐CH4 eddy covariance network (UpCH4 v1.0): Model development, network assessment, and budget comparison. AGU Advances, 4(5). https://doi.org/10.1029/2023av000956*

I led the analysis and writing of the study referenced above which was published in October 2023. I am aware of the ethical considerations regarding promotion of one's own work during the review process, but I would like to explain why I consider referencing of my study as not merely justified, but essential.

We thank you for reading the manuscript, providing suggestions to improve it, and the positive feedback.

Most of the work on this paper was carried out in 2022 and early 2023, with completion of the submission then being delayed for various reasons. This explains, but does not excuse, the fact that we do not compare the results with UpCH4 study (McNicol, 2023) in this manuscript. We updated the citation r, and would like to state that it was an omission due to a combination of circumstances, and not a deliberate action.

We include some comparisons with UpCH4 along the manuscript, emphasizing the complementarity and differences of the two approaches. However, notice that, strictly speaking, UpCH4 is a data-driven empirical upscaling of wetland methane fluxes using machine learning techniques. SatWetCH4 is a model based on a process equation - extremely simplified - which is also data-driven in terms of its calibration

and input data, but is not an empirical model. The authors would be interested in a future collaboration for a deeper inter-comparison study that could certainly improve our understanding of both product estimates.

Point by point replies to the comments are listed below.

The line numbers given hereafter refer to the tracking-changes PDF manuscript. Changes in the manuscript are given here in violet.

Given the lack of a true global wetland methane emissions benchmark dataset, our understanding advances via careful model/product inter-comparison. The absence of a comparison to our 2023 study greatly reduces the potential insights that could be gained. For instance, Bernard et al. estimate a global annual total wetland methane source of around 70-86 TgCH4 y-1. This is remarkably low; much lower than the spread of bottom up and top down models within the Global Carbon Project methane budget ensemble or WetCHARTS. I think the estimates may be different due to Bernard's inclusion of an explicit substrate availability term in the empirical function, over the temperature-dominated learned function in McNicol et al., or the inflexibility of the temperature response term (see section below), but this needs to be explored in some detail by the authors before a clear insight can be gained into the pros and cons of using this substrate dependent formulation.

The value of the SatWetCH4 total budget found by the independent calibration method presented in this manuscript is indeed remarkably low compared to the literature. We mentioned this several times in the manuscript, notably section 3.3.4.

*"SatWetCH4 is in the **lower** range of the GMB LSMs (grey areas), or even slightly below this range in the 30°S-30°N band. The total annual budget of SatWetCH4 wetland emission estimate averages 85.6Tg CH4 yr−1 with WAD2M (resp. 70.3 with TOPMODEL), which is **lower** than the range of the GMB LSMs estimates (102 to 181 Tg CH4 yr−1). This discrepancy is consistent with **the underestimation of methane fluxes by SatWetCH4** at tropical sites (discussed in section 3.2). Note that this difference in total emissions could be easily resolved by calibrating the k parameter to the total emissions of the mean GMB LSMs if we need to constrain total emissions, as it has been done previously by Bloom et al. (2017); Gedney et al. (2019)."*

We explained in the submitted manuscript that the global total estimate is not necessarily the value of interest in  this approach, but rather  the seasonal and inter-annual and spatial patterns. In fact, to be consistent with the literature mean global estimates, as we already mentioned in this manuscript,  we could simply calibrate the k factor to match the commonly accepted total, as it has been done in several previous studies (e.g., Bloom et al. 2017). Even though the total amount diverges from the estimates established in the literature, we believe that this kind of independent process-based model can be  useful for the community: the equation used here is a simplified process model, but with the advantage of being calibrated and constrained with independent data. We believe that the advantage of SatWetCH4 is its dependency on remote sensing observations rather than on modelling outputs, which allows looking at seasonal and inter-annual variations and spatial patterns.

The global total differences in methane emissions between UpcH4 and SatWetCH4 are due :
1. to the different nature of the models. Even though they are calibrated using the same eddy covariance data, UpCH4 is based on a data-driven empirical approach, while SatWetCH4 uses a simplified process-based equation.
2. SatWetCH4 tends to underestimate fluxes in the tropics due to the regression process used to optimize the parameters of the equation.

3. UpCH4 simulated very high values in Australia and subequatorial Africa (Sahel) compared to SatWetCH4 (and GMB), as shown in Fig. 5 of McNicol et al. (2023) and on Fig.6 of our revised manuscript. UpCH4 fluxes over Australia represent 20.5 TgCH4 yr-1 (14% of the global total) which seems huge knowing that these regions are a desert or semi-arid area. These high methane emissions over Australia result from the use of the WAD2M product for constraining the wetland extent, and its known discrepancies. WAD2M is based on the SWAMPS dataset (Jensen et al., 2019). In fact, the microwave signature of deserts has similar characteristics to water, and the SWAMPS dataset misidentifies desert, snow, and coastal regions as flooded areas (see Sect. 3.3.1 of the manuscript, Pham-Duc et al., 2017, Bernard et al., 2024). The same misinterpretation occurs over the Sahel, with emissions of 13.8 TgCH4 yr-1. Thus, these two regions alone explain 34 TgCH4 yr-1 of the 60 TgCH4 yr-1 difference between SatWetCH4 and UpCH4.

We can still discuss the low value of our global total estimate using SatWetCH4, as there are several possible reasons for it. Firstly, this extremely simplified equation does not take into account all processes (oxidation and transport to the atmosphere) and their related parameters (such as a water proxy, vegetation type). Soil moisture is indeed relevant at the site scale and improves the predictions at site-level. However, we found that available soil moisture or water table products do not provide any pertinent information when taken at a coarse resolution of 0.25°. So we decided to exclude this for the moment. This information, which is often used in land surface models (Wania et al., 2013) on a scale of 0.5° or 1°, does not in fact add any benefit at that scale. Another reason could be the underrepresentation of tropical sites, which have generally larger fluxes than temperate and boreal sites. More data in the tropic areas, as underlined in McNicol et al. (2023), would help to better represent tropical wetland emissions, and in the case of SatWetCH4 could allow a specific fit per latitudinal band, or per wetland class. Finally, the site level calibration method can also lead to underestimation, as the site fluxes are measured all year long, even if the site is unsaturated, and this can lead to an underestimation of flux intensity at the global scale. This is due to the fact that dynamic wetland mapping products account for saturated or inundated areas, whereas site-level measurements conducted during the dry season are likely to underrepresent the emission intensity of saturated areas. Consequently, the parameters calibrated from dry season measurements may underestimate emission intensity when multiplied by the area of saturated wetlands.

These points are deeper discussed in lines 366-381 of the tracking changes manuscript.

*"Figure 7 shows the latitudinal distribution per season for SatWetCH4 run with WAD2M and TOPMODEL, as well as the GMB LSMs, WetCHARTs ensemble, and UpCH4 estimates. The monthly variation for emissions estimates and wetland extent per latitudinal band is shown in Fig. 8. Note that the WetCHARTs models are calibrated to the GMB annual budget and are therefore not independent in terms of methane emission amplitude. SatWetCH4 is in the lower range of the GMB LSMs (grey areas), or even slightly below this range in the 30°S-30°N band. The total annual budget of SatWetCH4 wetland emission estimate averages 85.6 Tg CH4 yr−1 with WAD2M (resp.*

70.3 with TOPMODEL), which is lower than the range of the GMB LSMs estimates (102 to 181 Tg CH4 yr−1) and the UpCH4 estimates (146 Tg CH4 yr−1 ) even the same wetland extent is used.

This discrepancy can be explained by 1. an underestimation of methane fluxes by SatWetCH4 especially of tropical fluxes (discussed in Sect. 3.2 and in the following paragraph) and 2. the consideration by WAD2M of desert regions as inundated areas, leading to methane fluxes overestimation in Australia and Sahel in UpCH4 and some diagnostic LSMs (see discussion Sect. 3.3.3, Fig. 6, S3 and 7). Indeed, Sahel and Australia represent 33.4 out of the 146 Tg CH4 yr−1 estimated by UpCH4 using WAD2M, while these regions represent 4.5 Tg CH4 yr−1 in SatWetCH4 using WAD2M.

The scarcity of site-level data in tropical regions, coupled with the absence of tropical peatlands and floodplain sites, has undoubtedly contributed to the uncertainty associated with the calibration of parameters. Furthermore, the use of site-level calibration for tropical wetland emission may result in an underestimation at the regional or global scale. This is due to the fact that dynamic wetland mapping products account for saturated or inundated areas, whereas site-level measurements conducted during the dry season are likely to underrepresent the emission intensity of saturated areas. Consequently, the parameters calibrated from dry season measurements may underestimate emission intensity when multiplied by the area of saturated wetlands. This is a less significant issue in temperate and Arctic regions, where the wet seasons occur in summer and there is minimal emission in winter. As the number of tropical sites increases, future studies could consider refining the calibration for the tropics, for example, by only using wet season measurements for calibration.

Note that this difference in total emissions could be easily resolved by calibrating the k parameter to the total emissions of the mean GMB LSMs if we need to constrain total emissions, as it has been done previously by Bloom et al. (2017); Gedney et al. (2019)."

Another key distinction that would be valuable to explore via inter-comparison with the UpCH4 product is the choice to use an empirically defined model itself, rather than a purely data-driven (random forest) model as in UpCH4. As a community we are being encouraged to combine our ecosystem science knowledge of good process representation with a full utilization of flux observations, enabled with machine learning. I imagine this study could become a useful baseline study of a purely empirical model, which arrives at one calibrated parameter set (one model) from optimization on all ground surface methane flux data, much as UpCH4 is intended as a baseline for a purely data driven model, which optimizes on the same dataset to identify, in effect, an ensemble of highly conditional predictive models. Neither method is anywhere close to perfect, and future advances are most likely in uniting the two, yet Bernard et al. make no mention of this modeling issue within their limitations section. The behavior of the purely data driven models is of particular concern during expansive extrapolations where models may behave in an unconstrained (by data) manner, yet in McNicol et al. 2023, we arrived at a more plausible global total of 146 TgCH4 y-1. A discussion of model equifinality and extrapolation is entirely absent and would be readily facilitated by comparison to our published study.

We added some discussion about the complementarity of the two approaches (lines 477-487 of the tracking changes manuscript):

*Despite the impossibility of analyzing temporal variation due to WAD2M issues, Fig. 9 informs us that the temporal variations of SatWetCH4 are more similar to GMB LSMs than UpCH4. This is consistent with the fact that SatWetCH4 is a - highly simplified - process-based equation, whereas UpCH4 relies on empirical flux upscaling using random forest. SatWetCH4 and UpCH4 approaches both provide new independent estimates of wetland emissions, while offering distinct perspectives. A deeper comparison of the fluxes modelled by SatWetCH4 and UpCH4 at the site level could serve understanding differences between the simplification of complex processes represented by a fixed process equation (SatWetCH4) versus a machine learning data-driven approach (UpCH4). In addition, running both SatWetCH4 and UpCH4 with another wetland extent database would also serve to assess uncertainties and errors associated with WAD2M product and a better comparison of global methane emissions trends estimated by SatWetCH4 and UpCH4. Both methods are currently limited by the scarcity of eddy covariance flux data (McNicol et al., 2023), especially over important wetland methane emitting regions of the world, e.g., in the tropics (Congo, Sudd, Amazon) and Russia (Siberian lowlands).*

A more specific concern not considered by the present study related to this issue of empirical model calibration is that recent work has demonstrated that the temperate dependency of methane flux varies in space and time, such that a single temperature dependency is almost certainly an erroneous model framework assumption (Chang et al. 2021; Yuan et al. 2024). This is not addressed at all in the present study, nor are these two recent and high profile paper cited.

*Chang, K.-Y., Riley, W. J., Knox, S. H., Jackson, R. B., McNicol, G., Poulter, B., et al. (2021). Substantial hysteresis in emergent temperature sensitivity of global wetland CH4 emissions. Nature Communications, 12(1), 2266. https://doi.org/10.1038/s41467-021-22452-1*

*Yuan, K., Li, F., McNicol, G., Chen, M., Hoyt, A., Knox, S., et al. (2024). Boreal-Arctic wetland methane emissions modulated by warming and vegetation activity. Nature Climate Change, 14(3), 282–288. https://doi.org/10.1038/s41558-024-01933-3*

In our study, the $Q_{10}(T)$ value is not constant but depends on temperature ($Q_{10}(T) = Q_{0\ 10}^{T0\ /T}$). Ideally, we could adjust $Q_{10}$ to better match the observations, which would indeed be an improvement. Nonetheless, it is clear that $Q_{10}$ is one of, but not the primary, sources of uncertainty in our results. We conducted tests by latitude band or wetland types to explore this further. Unfortunately, fitting $Q_{10}$ on such groups is not reliable due to currently insufficient data.

We added some discussion about the $Q_{10}(T)$ simplification in the discussion lines 466 and 474-476, along with citation of the suggested literature:

*"Indeed, Q10 was found to depend on ecosystems (Chang et al., 2021)"*

*"Some refinement of the Q10 function (here $Q_{10}(T) = Q^0_{10}{}^{T0/T}$ according to Gedney (2004)) could be envisioned, such as the incorporation of a temperature hysteresis (Chang et al., 2021)."*

I am also concerned that the concept, structural elements, and visualization choices are remarkably similar to our 2023 study, despite no citation being present.

Although we agree that we should have cited your study and compared the results of SatWetCH4 with UpCH4, we disagree with this comment. Indeed, we were surprised by this remark implying a copy of your work.

First, as explained in the preamble to this document, almost all the study and the Figures were produced in 2022 / first part of 2023 before the publication of your paper.

The present SatWetCH4 study was also presented in June 2023 at the NCGG9 conference. If needed, we can provide versioning of the scripts used and of manuscripts, even if this suspicion is absurd given the classic structure of our manuscript and of the Figures.

Out of the ten figures in our study, only two are similar to yours: the flux towers localization maps and the flux visualization map. These are standard figures also used in several papers, e.g., the Global Methane Budget (Saunois et al., 2020) or FluxNet-CH4 paper (Delwiche et al., 2021) as shown below.

[Figure]

Figure 1 of our manuscript

[Figure]

Figure 2 of McNicol et al., 2023

[Figure]

Figure 3.a of Delwiche et al. (2021)

[Figure]

Figure 6 of our manuscript

[Figure]

Figure 1 of McNicol et al. (2023)

[Figure]

Figure 3 of Global Methane Budget, Saunois et al., 2020.

In addition to our project leads, our study included over 50 co-authors to honor the terms and spirit of a data policy agreement designed to provide fair credit to a large and growing international community of eddy covariance scientists. I was heavily involved in the acquisition of data in FLUXNET-CH4 v1.0 and a common concern voiced by international investigators outside Europe and North America, whose contributions would do much to address the data and community biases present across the global contributor network, was that their data may be used in high profile global synthesis studies for which they would not receive fair credit. While Bernard et al. do cite Delwiche et al. 2021, which is the dataset release paper, citing our study would ensure that the spirit of the data policy of the original FLUXNET-CH4 synthesis was not undermined by omission on downstream FLUXNET-CH4 upscaling work. The authors should also include, as indicated on the FLUXNET website, the second of the following to CC-BY data attribution requirements to include the DOIs of the sites contributing data:

***Data use should follow these attribution guidelines for CC-BY-4.0****:*

- Cite the data-collection paper, for example, for FLUXNET2015, cite Pastorello et al. 2020[1]
- *List each site used by its FLUXNET ID and/or per-site DOIs in the paper (these DOIs are provided with download)*

*Fluxnet.org/data/data-policy/ accessed July 6, 2024*

We are fully aware of and grateful for the collaborative efforts of all the PIs of the EC Flux Tower and have acknowledged them in the Acknowledgements section:
*Acknowledgements. [...] The authors would like to thank the managers of the 55 out of 58 eddy covariance flux towers who made their data available as open source. These data play a crucial role in improving our understanding of methane emissions from wetlands and in calibrating the models more accurately.*

We have tried to follow the legacy statement carefully by 1. citing the reference papers of the databases we used and 2. citing all DOIs of the 58 EC flux tower data used in the supplementary material attached to the manuscript (Table S2). We only used CC-BY-4.0 data from the FluxNET-CH4 and AmeriFlux datasets. For Euroflux, as some sites were not under CC-BY-4.0, we contacted the site PI to ask for permission to use their data and proposed co-authorship for non-legacy sites.

We believe that we are following the legacy terms fairly. We understand why you included all sites PI in your paper, as your upscaling approach relies heavily on FluxNET-CH4 fluxes, which is not the case in your study.

---

## Author Comment (AC3)

**Replies to Referee 3**

Juliette Bernard et al. developed a simple empirical model (SatWetCH4) to simulate global wetland methane emissions by leveraging large-scale remote sensing data. This model addresses uncertainties in wetland methane emissions and proposes a new approach for estimating substrate availability using MODIS data. Calibrated with eddy covariance flux data, the model aims to provide a simpler, faster alternative to more complex Land Surface Models. While I appreciate the authors' development of this new model and its contribution to global wetland CH4 monitoring, I have several major comments and concerns:

We thank you for reading the manuscript and your comments. Hereafter are point by point replies to your concerns.

The line numbers given hereafter refer to the tracking-changes PDF manuscript. Changes in the manuscript are given here in violet.

Major Issues:

1. Novelty and Approach:

One of the novel aspects of this paper is directly modeling large-scale substrate availability for methanogens. It uses NPP and soil organic carbon turnover (Eqn. 2) to estimate the carbon substrate mass balance. However, I have two questions regarding this approach:

Methanogens use CO2 or acetate as substrates. What exactly does C_substrate represent? Is there evidence that the modeled C_substrate correlates with actual substrates for methanogens, making it a valid proxy variable? In other remote-sensing-based approaches (e.g., Bloom's WetCHARTS), soil respiration rate (CO2 flux) is used as a proxy for substrate availability.

Methanogens are using low molecular weight soil organic compounds which are easily available: acetate, CO2 or methylated substrates (Torres-Alvarado et al., 2005; Nzotungicimpaye et al., 2021). This available fraction of soil organic carbon (SOC) is not a measurable quantity, and a proxy has to be determined for methane emissions modelling purposes. The available SOC for methanogenesis is the remaining altered fraction of biopolymers produced by living organisms through photosynthesis, and methanogenesis has been shown to be correlated with plant productivity (NPP, GPP) (e.g., Whiting et al., 1993; Updegraff et al., 2001, Knox et al. 2021).

In the literature, three main approaches/proxies are used in bottom-up models for methanogenesis:

- the simplest approach is indeed suggesting that methane emissions are a fraction of CO2 emissions by microbial decomposition (heterotrophic respiration) (e.g., WetCHARTs, LPJ model).
- some models use directly plant productivity NPP (or GPP) as proxy for available SOC (e.g., Walter et al., 2000, UW-Vic model)
- other approaches relies on simulated carbon pools (e.g., ORCHIDEE model)

Here we suggest a novel approach employing a fusion of the two last methods, a one-pool SOC model relying on NPP remote-sensing data for organic carbon input.

WetCHARTs, as our model, is a one-step approach model representing implicitly methane production and directly estimating methane emissions. Bloom et al. are using remote sensing products only to control the dynamic part of the wetland fraction, and not the SOC source. In WetCHARTs, methane emissions depend on heterotrophic respiration estimated from terrestrial biosphere models, consequently is not an independent approach. The heterotrophic respiration does not only represent available carbon but is a complex function, embedding dependence to other variables such as temperature.

How is C_substrate validated at sites in terms of its control over wetland CH4 emissions? Does eddy covariance site data show a strong relationship?

Validation Concerns:

The validation of C_substrate (Figure 5) is not convincing for two reasons. SoilGrids and HWSD provide benchmarks for upland soil carbon stock but not wetland C_substrate. Additionally, HWSD/SoilGrids provide total soil carbon stock, which does not necessarily turn over quickly. However, the defined residence time of C_substrate is less than 5.5 years (section 2.1).

Even the substrate availability plays a major role in methanogenesis, this is not a measurable quantity. We explicitly write in the submitted manuscript that HWSD and SoilGrids do not represent the same quantity as $C_{substrate}$, as these products represent total soil organic carbon (lines 274-279 cited below).

*"The 2003-2020 mean map of the Csubstrate product is shown in Fig.5. This product is used as a representation of the soil carbon substrate available for methanogenesis. It should be noted that there are no analogous products for evaluation. We suggest a comparison with global estimates of 0-100cm SOC stocks derived from the World Soil Database (HWSD) (Wieder, 2014) and SoilGrids (Hengl et al., 2017) to see differences between our proxy for available substrate compared to total organic carbon stocks. The latitudinal distribution and the latitudinal distribution normalized by the latitudinal maximum of the three products are shown on the right side of the figure."*

A comparison of this $C_{substrate}$ to total organic carbon stock is discussed, as no other dataset of carbon substrate is available. The soil carbon substrate $C_{substrate}$ variations present low temporal variations, and constrain the emissions spatial pattern in the model rather than the temporal variations (lines 287-293). Not taking into account a carbon availability parameter would be tantamount to assuming that SOC availability does not affect methane production/emission, which is false.

If the reviewer knows of another dataset to compare $C_{substrate}$ with, we would be happy to use it.

2. Q10 Parameter:

Q10 is a key parameter in the proposed modeling approach. However, Q10 is a complicated variable for wetland methane emissions due to different temperature sensitivities for various methanogens and methanotrophs. The emergent Q10 has been found to be small when constrained by satellite CH4 concentrations and inversions (Shuang Ma et al., 2021). This work uses a simply calibrated value of Q10 (2.99). I suggest a more in-depth discussion of the high Q10 value in existing literature, including reasons for discrepancies, implications, and biogeochemical processes.

Bloom et al. (2017) tested in WetWHARTs ensemble 3 fixed Q10 values : 1, 2 and 3. Shuang Ma et al. (2021) found that WetCHARTs outputs with Q10 = 1 were better agreeing with top-down inversion approaches. However, WetCHARTs equation includes heterotrophic respiration (Bloom et al., 2017):

$$F(t,x) = s\,A(t,x)\,R(t,x)\,q_{10}^{\frac{T(t,x)}{10}}, \qquad (1)$$

where A(t, x) is the wetland extent fraction, R(t, x) is the C heterotrophic respiration per unit area at time t, and T (t, x) is the surface skin temperature) and s is a global scaling factor. Their Q10 then represents "the temperature dependence of the ratio of C respired as CH4 (where Q10 is the relative CH4 : C respiration for a 10 ∘C increase)", **and not the methane production dependence to temperature**. A temperature dependency is already embedded in the heterotrophic respiration term R. WetCHARTs Q10, named hereafter $Q10_{CH4:C}$ and our Q10, named hereafter $Q10_{CH4}$, are then different. The results from Shuang Ma et al. (2021) showed best emissions fitting with $Q10_{CH4:C}$=1 suggesting that using the temperature dependence for methane emissions already embedded in the heterotrophic respiration function R(t, x) is better than increasing the temperature dependency by adding the control of the second $Q10_{CH4:C}$, i.e., $Q10_{CH4:C}$ value >1.

In our case, having a $Q10_{CH4}$=1 in our equation would imply that temperature has no influence on methane emissions, which is absurd as temperature has been extensively documented to be a major driver of methane emissions variations (e.g., Know et al., 2021, Delwiche et al., 2021, Kuhn et al., 2022), at least in northern and temperate sites. Our $Q10_{CH4}$ value is in the range of other model values found in literature for methanogenesis which has been extensively discussed in Section 3.1 of our manuscript (lines 210-226):

[Figure]

*Figure 2*
*"Nzotungicimpaye et al. (2021) in WETMETH proposed a Q10 (T) formulation such that, when incorporated into the equation Q10 (T)$^{(T-T0)/10}$, it indicates an optimal temperature range for methanogenesis around 25-30°C.*

*Although we attempted a similar approach to formulate Q10 (T), it resulted in minimal changes in the flux outcomes while increasing the complexity of the formulation and hindering the convergence of the cost function. Albuhaisi et al. (2023) used a reduced Q10 for temperatures above 5°C or above 30°C, resulting in abrupt transitions at these temperature thresholds. However, this implementation may not be appropriate for global analyses, as tropical wetlands experience temperatures above 30°C, and such sudden changes do not reflect of physical reality. Therefore, the Gedney (2004) formulation $Q10 (T) = Q0_{10,opt\,0}$ was used for calibration, resulting in Q10(T) from 3.12 (-10°C) to 2.60 (40°C), which is slightly lower than the Gedney (2004) value (3.89 at -10°C to 3.13 at 40°C). Our Q10(T) value contrasts with that of Walter and Heimann (2000) (Q10 = 6.0, no temperature dependence), but closely matches the value chosen by Albuhaisi et al. (2023) for the 5°C-30°C range (Q10 = 3.1 for T between 5°C and 30°C, Q10 = 2.0 below 5°C or above 30°C). Consequently, similar $Q10(T)^{(T-T0)/10}$ curves are observed in Fig.2.b between our estimate and those of Gedney (2004) and the 5-30°C range of Albuhaisi et al. (2023), although our formulation exhibits slightly lower values. This would result in a slightly lower increase in methane fluxes with soil temperature."*

We add a sentence (lines 226-228) for comparison with in situ data :
*"The Q10(T) found in this study is also in agreement with meta-analysis of Q10 defined from in situ data, e.g., 2.8 in Kuhn et al. (2021) and 2.57 in Delwiche et al. (2021)."*

Additionally, we removed WetCHARTs Q10 values from the comparison as it does not represent a $Q10_{CH4}$ but a $Q10_{CH4:C}$, as discussed above.

3. Model Simplicity and Missing Processes:

While I appreciate the simplicity of the model equation, capturing the major dynamics of wetland CH4 emissions, some important processes are not represented. Vegetation phenology (Carole Helfter et al.), which has remote sensing data available (EVI or NDVI), and atmospheric pressure, which controls the bubbling processes of wetland CH4, are significant. Including or at least discussing these variables/processes in the next version would be valuable.

This study indeed develops a simple model equation based on observations, capturing first order variations in methane production to temperature, wetland fraction and carbon substrate, that will serve in future studies to investigate time variations over long time periods and methane emissions trends.

In the near future, we would like that vegetation could be taken into account in a further version of SatWetCH4. However, in the present work we discussed two possibilities in Sect. 4:

- using factor/formulations for transport processes as modeled by LSMs. We excluded this approach because we want SatWetCH4 to provide CH4 emissions independently of the modeling parameters/outputs of LSMs.
- or taking into account implicitly these transport processes and vegetation dependency by fitting the equation parameters per vegetation or wetland type. This approach has been tested and discussed in the manuscript. Indeed, with only few sites per wetland type, it is difficult at the moment to calibrate the model per wetland type. More data in upcoming years would certainly enable future simple models to take transport processes and vegetation dependency into account. This is discussed in lines 456-474 of the manuscript.

*"The simplified SatWetCH4 model we have developed makes important approximations that imply important shortcuts. In particular, no distinction is made between methane production and emissions. This supposes that SatWetCH4 one-step equation includes production, oxidation, and transport in a single formulation, which are sometimes distinguished in some of the more complex LSMs (Wania et al., 2013; Morel et al., 2019; Salmon et al., 2022). Among the 3 pathways of methane transport in wetlands, including diffusion, ebullition and plant-mediated transport, plant-mediated transport is the dominant one (Ge et al., 2024). Ge et al. (2024) have recently published a comprehensive review of the role of plants in methane fluxes, showing their influence not only on methane transport but also on methane production and oxidation. Feron et al. (2024) also show that trends in methane flux changes at the site level depend on ecosystem and vegetation type. Accounting for the different vegetation classes therefore appears to be a possible improvement to our simplified approach.*

*A simple way to account for this in the SatWetCH4 model at a first order would be to fit the scaling factor k and/or Q010 as a function of vegetation class or wetland type. We performed such calibration tests, taking into account the wetland classification. However, the cost function either did not converge due to the small number of sites per category, or the result was highly dependent on few sites, thus overfitting results. In fact, eddy covariance flux towers measuring methane emissions are not evenly distributed around the globe and their distribution is highly skewed, as discussed in part 2.2. Some wetland categories are poorly represented, for example, there are only two mangrove sites. This scarcity of data makes this type of calibration highly uncertain. However, we can expect an improvement in the coming years, as in situ methane measurement is a rapidly growing field, as shown by the increasing number of flux towers along the years in the Supplementary Table S1. Future data, especially in the tropics, will be essential to better constrain the models and to include more processes into account."*

and in the conclusion (lines 494-499):

*At the site level, the SatWetCH4 model reproduces well the boreal fluxes and most of the temperate fluxes, but poorly the emissions seasonality of the tropical sites. This could possibly be improved in future studies by adding high resolution information on local water availability (SWC). Another important improvement would be a calibration per wetland type, which would allow the influence of vegetation to be taken into account as major transport pathways. For this, more eddy covariance flux measurements in the tropics are essential to gain a deeper insight into the processes governing temporal variations in this latitudinal band, and to develop and calibrate this one-step model.*

Atmospheric pressure is a predictor at multiday scale, but not at monthly ones (Knox et al., 2021).

4. Validation Performance:

Related to comment 4, the performance of SatWetCH4 at validation sites (Figure 3) is not satisfactory. This suggests that temperature and C_substrate alone may not be sufficient to capture the observed CH4 emission variability at the scale of eddy covariance sites. More effective calibration or the inclusion of missing dominant control variables in Eq. 1 may be necessary.

As discussed in comment 3, SatWetCH4 is a simplified model which, despite its simplicity, can estimate CH4 emissions as well as more complex LSMs and could be useful later for sensitivity studies, since it is constrained by observations. Only few flux data are available to constrain methane emissions models and accurately simulate methane emissions spatio-temporal variation. In the case of methane emissions, dominant control variables are numerous (temperature, available substrate, water, 3 types of transport processes, bacteria presence, etc.) and are formulated implicitly or explicitly in models which determine the level of model complexity. However, complex models ideally demand to be calibrated for each explicit process considered, which is not the case and result in increasing model uncertainties. Additionally, the level of the model complexity is usually determined based on the objective of the study, whether the study aims at understanding local processes or assessing global variability and trends. Here we wanted to develop an independent model, mainly constrained by satellite observations to study the methane fluxes spatial-temporal variability, therefore, we employed a simple model that is calibrated like other models using in situ site observations. Model limitations and missing processes, such as humidity or vegetation, are extensively discussed in the manuscript (Section 4, lines 419 to 487).

Other global models present the same level of uncertainties compared to site fluxes. It is for example interesting to note that data-driven machine learning approach from McNicol et al. (2023) has similar error values than SatWetCH4 (see Figure 3 of McNicol et al., 2023), even though they considered the 6 dominant predictors of methane fluxes. They also claim that the future flux data will make it possible to better describe the fluxes of all types of wetlands around the world.

5. Global Emission Estimates:

SatWetCH4 extrapolates site parameters to global wetlands. When compared with other products (GCP or WETCHARTS), SatWetCH4's emissions appear significantly lower, particularly tropical emissions throughout the year and boreal/arctic emissions in June/July/August. This may be due to SatWetCH4's low bias at sites with high emissions (Figure 3C).

WetCHARTs is constrained to match the mean value of total annual budget of GMB (GCP) models estimates, therefore it is not an independent estimation. Land surface models and Top-Down estimates and are calibrated with previous estimates (GMB, Saunois et al., 2020). Moreover, there is no evidence to suggest that these models represent an absolute truth.

SatWetCH4 global emissions are, indeed, remarkably low compared to GMB. We could also have calibrated the model to match other model estimates, but we think it is more interesting to present this independent approach. SatWetCH4 provides an independent estimates that is compared to most recent literature references work, and we did not claim that SatWetCH4 total value is more reliable than GMB estimates. The SatWetCH4 model is rather intended to study inter-annual variations based on independent remote sensing observational products. This low value was extensively discussed throughout the manuscript and more specifically in the section 3.3.4 which has been extended with a comparison to UpCH4 (McNicol et al. 2023) (see lines 360 to 381).

6. Summary

In summary, while I appreciate the authors' attempt to model global wetland CH4 emissions, I must point out that major variables and processes are missing in SatWetCH4. The site-level model calibration is not effective, and the global estimate of wetland CH4 emissions is significantly lower than values in the literature. I look forward to seeing an improved version during revision.

The purpose of this study is to develop SatWetCH4, which is indeed a simplified one-step model of methane emissions. A model is necessarily a simplification of reality. This simplification varies between approaches. Very complex LSMs try to represent many of the processes involved in methane emissions, but are then poorly constrained. We proposed here a simplified satellite-based approach, calibrated with available eddy covariance data. In a process-based approach, the processes can be represented explicitly or implicitly. Here we implicitly consider methane production, oxidation and transport by directly simulating methane emissions. We have discussed the advantages and limitations of such an approach in the manuscript and then in the responses to your comments.

Other variables, such as vegetation dependence, could be included in SatWetCH4 in the future as we expect new flux tower data to be available. Running the model at a finer scale would also help to account for variations in soil humidity, but at the moment wetland fraction datasets are not available at a finer resolution. The strengths of SatWetCH4 are to provide a new, independent approach, largely based on satellite data, and to offer the potential to study CH4 flux inter-annual variations.

Despite its simplicity, SatWetCH4 is as good (or as bad) as other more complex global models.

We hope that this improved version of the manuscript and these exchanges will clarify the aim of our study.

---

## Author Response (AR3)

**Replies to the editor**:

Dear Dr. Bernard,

I have read your response letter and the two rounds of comments by the reviewers. As you can see in the second round, both reviewers remain concerned about the ~50% underestimation of wetland emissions at high-emission sites and on a global scale. While I recognize that your model provides a new, independent estimate of wetland emissions, this issue requires further discussion. For example, how well does the fitting perform? Would it be possible to show a scatter plot of the observed versus modelled points and assess deviations from the 1:1 line? Additionally, if you are able to tune the parameters to match the global budget, would it bias the site-level estimates?

I would appreciate it if you could provide further details on these points.

Best regards,
Yilong

Dear Yilong,

Thank you very much for reading the exchanges of the review process and for your suggestions.

To address the issue raised by the reviewers and following your advice, we have made a scatter plot of the observed and modeled fluxes at site levels, where the colorbar represents the density of the measurements, as it is most commonly done, with a 1:1 line (dashed dark line) and the linear regression through the origin (0,0) (dashed blue line). This Figure shows that the model is slightly biased toward lower values when comparing to monthly site data. We have included this Figure in the Supplementary as Figure S1. Regarding the tuning of k and Q10 to match other global budget estimates (e.g. GMB), 2 parameters (k and Q10) are sufficient to achieve both a constraint on the monthly site data and a constraint on the global estimates, with similarly low biases on the monthly estimate scatter plot. However, this estimate would then lose its independence from the others estimates.

[Figure]

Indeed, the total output of SatWetCH4 in this manuscript (86TgCH4/yr) is below the bottom-up estimates, e.g., estimates of the Land Surface Models (102-182TgCH4/yr) or McNicol et al. 2023 (103-189 TgCH4/yr). SatWetCH4 certainly underestimates global methane emissions. Though, we would like to remind here that the LSMs have been generally calibrated against top-down estimates or other historical values and that such calibrations lack independence across LSMs.

SatWetCH4 is a simplified model, with the aim of running it on a global scale, using satellite observations as input data to provide independent estimates from top-down or LSMs estimates. While the model's emissions estimate is lower than the GMB LSMs estimates, it remains within a comparable range and captures similar spatio-temporal variations, making it suitable for further studies. We also show in the manuscript that adding more variables at 0.25° resolution (such as WTD or SWC) with currently available datasets does not improve model accuracy. This issue of the local scale to large scale representativity is never challenged in the global models' design, which extrapolate relationships observed at the site level.

This manuscript, submitted to Global Model Development, aims to describe the approach of SatWetCH4. **We believe that the value and strength of this method does not lie in providing a precise total global wetland methane budget, but in offering a straightforward and efficient tool for investigating large-scale spatio-temporal changes in wetland emissions.** We are currently carrying out two studies based on the development presented in this manuscript.

1. The first study is to **run SatWetCH4 using satellite data to examine large changes** in terms of inter-annual variations, long-term trends, and spatial patterns that could be explained by changes in temperature and wetland extent. The aim is to determine if such a simple approach can explain the spatial pattern of wetland methane emissions and the recent atmospheric methane changes over the last decades in terms of concentration and isotopic signature. This was not possible due to the limitations of WAD2M, but we have recently derived a new dataset of wetland extent (in review in ESSD https://doi.org/10.5194/essd-2024-466) that allows us to model and study wetland methane emissions over 30 years.

2. Another study aims also to **calibrate SatWetCH4 using an inversion model.** Here, we calibrated the two parameters k and Q10 in using in situ measurements, which are limited (58 sites), and really sparse over the Tropics. The use of an atmospheric inversion framework will allow optimizing the k and Q10 with global satellite data of methane concentration (e.g., GOSAT data). This approach will also provide observational constraints on total methane emissions and sinks. As more data with a global coverage will be available, the optimization of Q10 (and k) could be refined and performed per latitudinal band or wetland types (provided by a prescribed map such as GLWDv2).

We have revised the manuscript, particularly in the conclusion, to better clarify our objectives, which we believe are now more apparent. While we feel the manuscript clearly conveys its goals, we remain open to further revisions if the editors or reviewers suggest alternative wording.

Best regards,

Dear Dr. Bernard,

Thank you for your response letter. The review's report 1 raised the concern that SatWetCH4 underestimate at high-emission sites (years). It is indeed visible in your revised Figure S1, with a slope of only 0.8, indicating that you model underestimate the emissions by about 20%. Why the calibration did not achieve a close fit near the 1:1 line? This is also the reviewer's concern whether the parameterization or model structure is too simplified.

Best regards,

Yilong

Dear Yilong,

thanks for your reply and raising concerns.

Indeed, this does not match the 1:1 (regression through 0,0 gives 1:0.83) with individual monthly data for 2 reasons :

1) the cost function does not imply a linear regression through (0,0).

2) also, and more importantly, we weighted the cost function to give different weights to the monthly data. These weights are described in lines 197-205 of the manuscript. We did this because some sites have much longer time series than others. In particular, boreal and temperate sites have longer time series than tropical sites. Giving the same weight to each monthly data would therefore lead to an even worse under-representation of tropical sites. This is why the calibration was not done by a simple regression on all monthly data.

[Figure]

Least squares regression is performed simultaneously on all sites using the Broyden-Fletcher-Goldfarb-Shanno algorithm (Byrd et al., 1995). For sites with less than 12 months of data, a weight proportional to the number of monthly measurements is assigned to the site data. Sites with more than 12 months of data are given equal weights. The minimized cost function is :

$$\quad J = \sum_{sites} w_{site} MSD_{site} = \sum_{sites} w_{site} \overline{(F_{CH_4 obs} - F_{CH_4 sim})^2_{site}} \tag{3}$$

where $w_{site}$ is the site weight, $MSD$ is the Mean Square Deviation, $F_{CH_4 obs}$ is the in situ methane flux observed at the sites, and $F_{CH_4 sim}$ is the methane fluxes simulated by the model. If the number of monthly methane flux measurements at the site, $n_{site}$, is greater than or equal to 12, $w_{site} = 1$ otherwise $w_{site} = \frac{n_{site}}{12}$. Different initial parameter sets for $k_{firstguess}$ (0.01, 0.1, 1, and 10) and $Q^0_{10 firstguess}$ (1.5, 2.5, 3, and 4) are tested to evaluate the influence of the calibration initialization
205 and to ensure the global nature of the found minimum.

We hope this clarifies and justifies why the regression does not match the 1:1 of monthly data.

We respond to the reviewer concerns below.

Best regards,

Report 1 (Referee 3):

Most of my comments were addressed. I appreciate the authors' efforts. However, my biggest concern is that the model is over-simplified so it couldn't capture the variability of CH4 flux across 58 EC sites. Figure 3c is a clear demonstration that the model significantly underestimates CH4 emission, at high-emission sites (years).

Given the model is not parameterized well at EC sites (or even not capable of being sufficiently parameterized due to over-simplification), SatWetCH4 wetland emission estimate became so low. I just couldn't be convinced that global wetland CH4 emission is only 85.6 Tg CH4 yr−1 when upscaled with WAD2M (or 70.3 when upscaled with TOPMODEL). Either improving parameterization of the model or updating the model structure is needed 1) to sufficiently capture the CH4 emission dynamics (Figure 3c), 2) to make reliable upscaling products at global wetlands.

Thank you for reviewing the modified manuscript and for your valuable feedback. Below are our responses to your concerns.

We acknowledge that the SatWetCH4 model output (86 Tg CH4/yr) is lower than current bottom-up estimates, such as those from GMB's Land Surface Models (102-182 Tg CH4/yr) or McNicol et al. (2023) (103-189 Tg CH4/yr). While SatWetCH4 certainly underestimates global methane emission, we would like to remind here that the LSMs have been generally calibrated against top-down estimates or other historical values, which introduces dependencies across models and approaches. Also LSM studies usually do not provide any assessment against flux tower measurements.

The objective of this manuscript is to present the SatWetCH4 approach. While the model's emissions estimate is lower than the GMB LSMs estimates, it remains within a comparable range and captures similar spatio-temporal variations, demonstrating its potential for large-scale analysis. We believe that the value and strength of this method does not lie in providing a precise total global wetland methane budget, but in offering a straightforward and efficient tool for investigating large-scale spatio-temporal changes in wetland emissions (detailed in the reply to the editor).

SatWetCH4 is a simplified model designed for global-scale use, utilizing satellite observations as input data to provide independent estimates from top-down and LSMs methods. As such, it is not intended to precisely replicate site-level observations, which would require more detailed input data at high resolution (e.g., Water Table Depth, as discussed in the manuscript). Multisite optimization is challenging, especially when sites from all latitudes are included. Indeed, most studies focus exclusively on boreal and/or temperate regions, where the seasonal cycle is more predictable. For example, the Figure 3 below from the global study by McNicol et al (2023) , shows similar discrepancies than SatWetCH4, especially at tropical sites (e.g., BW-Gum), despite using more complex machine learning-based upscaling approaches with a greater number of predictors. This can also be seen in Figure 7 of Salmon et al. (2022) below: multi-site optimization is challenging and leads to underestimating the most emitting sites and overestimating the least emitting sites, even when the model used is complex and only tested for temperate and boreal peatlands.

Finally, we also show in the manuscript that adding more variables at 0.25° resolution (such as WTD or SWC) does not improve model accuracy, even though we see relationships between these variables and methane fluxes at the local site scale. This highlights an

inherent challenge in global modeling: extrapolating relationships observed at site-level measurements to large-scale estimates. This issue of the local scale representativeness to large scale is often not challenged in global models' design, which extrapolate relationships observed at the site level.

The manuscript has been revised to better clarify our objectives, which we believe are now clearer. While we feel that the manuscript clearly communicates its aims, we remain open to further suggestions for alternative wording.

[Figure]

Part of Figure 3 from McNicol et al (2023). Random forest model predicted versus observed values for(a–d) the mean seasonal cycle (MSC) of methane (CH4) flux for sites in (a) tundra,

(b) boreal, (c) temperate, and (d) tropical climate regions [...] The 1:1 fit is shown as a dashed black line.

[Figure]

Figure 7 from Salmon et al. (2022): Differences in annual methane emissions defined between the observed data (Obs), and simulations employing parameters optimized by the single site (SS) and by multi-site (MS) approaches.

Report 2 (Referee 1):

While I appreciate the authors' efforts to address all the previous comments, I have the same concern raised by the other two reviewers regarding the underestimation of global methane emissions. This issue might confuse readers about the global budget and needs to be discussed in more detail. The authors suggested that the scaling factor k could be easily tuned to match the global values, but it remains unclear how this adjustment would impact the model's performance? For example, would such a correction improve or degrade the site-level results, and to what extent? I would be pleased to see further discussions on this issue.

Thank you for your review of the revised manuscript. Below is our response to the concerns you have raised.

We acknowledge that the SatWetCH4 model output (86 Tg CH4/yr) is lower than current bottom-up estimates, such as those from Global Methane Budget (GMB)'s Land Surface Models (102-182 Tg CH4/yr) or McNicol et al. (2023) (103-189 Tg CH4/yr). While SatWetCH4 certainly underestimates global methane emissions, we would like to remind here that the LSMs have been generally calibrated against top-down estimates or other historical values, which introduces dependencies across models and approaches. The objective of this manuscript is to present the SatWetCH4 approach. While the model's emissions estimate is lower than the GMB LSMs estimates, it remains within a comparable range and captures similar spatio-temporal variations, demonstrating its potential for large-scale analysis. We believe that the value and strength of this method does not lie in providing a precise total global wetland methane budget, but in offering a straightforward and efficient tool for investigating large-scale spatio-temporal changes in wetland emissions (see the details of these studies in the reply to the editor). We have revised the manuscript to better clarify the goal of SatWetCH4, and remain open to further suggestions for alternative wording.

Concerning the tuning of the model parameters to match other global budget estimates (e.g. GMB), this would be possible by modifying the cost function (eq 3) to also add a constraint on the global estimates. Having 2 parameters to adjust (k and Q10) is sufficient to obtain a regression with similarly small biases on the monthly estimate. However, this estimate would then lose its independence from the other estimates.